# ZenoPS: A Distributed Learning System Integrating Communication Efficiency and Security

**Cong Xie \*** , **Oluwasanmi Koyejo** and **Indranil Gupta**

Department of Computer Science, University of Illinois Urbana-Champaign, Urbana, IL 61801, USA;
sanmi@illinois.edu (O.K.); indy@illinois.edu (I.G.)
**\*** Correspondence: cx2@illinois.edu or cong.xie@bytedance.com

**Abstract:** Distributed machine learning is primarily motivated by the promise of increased computation power for accelerating training and mitigating privacy concerns. Unlike machine learning on a single device, distributed machine learning requires collaboration and communication among the devices. This creates several new challenges: (1) the heavy communication overhead can be a bottleneck that slows down the training, and (2) the unreliable communication and weaker control over the remote entities make the distributed system vulnerable to systematic failures and malicious attacks. This paper presents a variant of stochastic gradient descent (SGD) with improved communication efficiency and security in distributed environments. Our contributions include (1) a new technique called *error reset* to adapt both infrequent synchronization and message compression for communication reduction in both synchronous and asynchronous training, (2) new *score-based* approaches for validating the updates, and (3) integration with both error reset and score-based validation. The proposed system provides communication reduction, both synchronous and asynchronous training, Byzantine tolerance, and local privacy preservation. We evaluate our techniques both theoretically and empirically.

**Keywords:** distributed; SGD; communication; security; privacy

## 1. Introduction

Recent years have witnessed the increasing attention to distributed machine learning algorithms [1–4]. The motivation to train machine-learning models in a distributed manner arises from the rapid growth of the sizes of machine-learning models and datasets, the increase in the diversity of the datasets, and the privacy concerns of centralized training alternatives. On one hand, we can use multiple GPU devices to accelerate the training with more computation power. On the other hand, training machine-learning models on the local private data on the massive edge devices makes the models more informative and representative. However, in both cases, the communication overhead between the devices is the potential bottleneck of the performance. To make things worse, larger learning systems with more devices are also more vulnerable to software/hardware failures, as well as malicious (Byzantine) attacks.

In practice, different scenarios require different solutions. For example, when conducting training tasks in a traditional datacenter, we do not require too much reduction of the communication overhead, or too much asynchrony, since the communication is relatively fast. In such a scenario, the requirement of privacy preservation and Byzantine tolerance is lower. However, when there are remote devices that are geographically far away from each other, the system should still work well with more limited communication. Additionally, the system should also support an asynchronous mode, in case there are stragglers. Furthermore, for edge devices, the system has to prepare for unreliable or malicious entities and provide guarantees in preserving privacy for users of remote devices (e.g., local differential privacy in sending messages to the central servers). The upshot

of this is that any system must be flexible enough in order to fulfill the requirements of different scenarios.

Besides the system-level concerns, a machine-learning system also needs to guarantee the performance of model training. However, there are always trade-offs. Reducing the communication overhead usually causes performance regression due to the inaccuracy in training. Robustness or fault tolerance also causes additional noise in training. It is essential to develop algorithms that satisfy the system requirements while guarantee training performance by providing theoretical analysis to show the trade-offs.

An efficient, secure, and privacy-preserving distributed learning system benefits a wide range of real-world applications. For example, communication compression can reduce the time cost by training [5,6] for large-scale deep-learning models such as BERT [7] and GPT-2 [8]. Federated learning [9] (FL) is another application that potentially benefits from our proposed distributed learning system. FL is designed to train machine-learning models on a collection of remote agents with their local private datasets, where the communication is extremely slow or unreliable, thus requiring compression [10] and protection [11]. For example, next-word prediction is a widely used feature on virtual keyboards for touchscreen mobile devices, which is commonly supported by FL due to the privacy concerns. However, the mobile devices are not always connected to high-speed and free WiFi. To make things worse, nefarious users can easily feed poisoned data with abnormal behaviors to attack the learning system. In that case, a distributed learning system that integrates both communication efficiency and security is very useful. Industrial machine-learning applications such as the ML-embedded energy sector [12,13] can also use distributed learning systems to train a global model on the data placed on multiple sites that are remote to each other, without transmitting the local training data, which are potentially confidential. In such cases, the communication efficiency and robustness are substantial due to the slow and unreliable networking of the remote sites located in suburban or even offshore areas [14]. However, combining communication efficiency, security, and privacy preservation is challenging in both theory and practice.

In this paper, we study distributed stochastic gradient descent (SGD) and its variants, which are commonly used for training large-scale deep neural networks. We present a distributed learning system that trains machine-learning models with variants of distributed SGD, which integrates several techniques in (1) communication reduction, (2) asynchronous training, and (3) tolerance to Byzantine faults.

In contrast to previous works that focus on one of the aspects in communication reduction [15], asynchronous training [16], and Byzantine tolerance [17–19], this paper presents a distributed learning system, ZenoPS, with the following characteristics:

- **Communication efficiency and security in a single system:** In contrast to the previous works that focus on either communication efficiency or security in distributed SGD, this paper presents algorithms that achieve communication reduction with both message compression and infrequent synchronization, asynchronous training, and Byzantine tolerance, simultaneously.
- **Detached validation from the servers:** In this paper, we present a new system architecture with an additional component called the *Validator*, which decouples the Byzantine tolerance from the server side. By doing so, the servers can focus on maintaining the global model parameters, which requires less computation resources, while the validators, which are more computation-intensive, are tasked with defending the servers by verifying the anonymous updates.
- **Local differential privacy:** In this paper, we propose to randomly insert Byzantine failures on the workers intentionally to produce noisy updates for the protection of the private local data on the workers. Combined with the validators, the system achieves both local differential privacy with limited regression in training convergence and accuracy.

The contributions of this paper are as follows:

- We present a distributed learning system, ZenoPS, that integrates the techniques of communication compression, different synchronization modes, and Byzantine tolerance.
- We present a novel system design with implementation details of ZenoPS.
- We establish the theoretical guarantees for the convergence, Byzantine tolerance, and local differential privacy of the proposed system.
- We show that the integrated system can achieve both communication efficiency and security in the experiments.

The rest of this paper is organized as follows. In Section 2, we briefly discuss the previous research related to our work. Section 3 formalizes the distributed optimization problem solved in this paper, with the detailed definition of the parameter-server architecture, Byzantine failures in distributed SGD, and local differential privacy. In Section 4, we present the algorithm and the system design of ZenoPS. The theoretical analysis of the convergence, Byzantine tolerance, and privacy preservation can be found in Section 5, with detailed proofs in Appendix A. We present the empirical results in Section 6. Finally, we conclude the paper in Section 7.

## 2. Related Work

This paper leverages previous works providing communication reduction with error reset [15], asynchronous federated optimization [16], and Byzantine tolerance [17–21]. The authors of [15] presented a technique called *error reset*, which adapts arbitrary compressors to distributed SGD and corrects for local residual errors, but the proposed algorithm and the corresponding theoretical analysis are limited to the classic synchronous SGD rather than the federated optimization. The authors of [16] presented a combination of asynchronous training and federated optimization or local SGD, but it lacked guarantees in security and privacy. For the security, we focus on the Byzantine failures [22] in this paper. The authors of [17,20,21] presented Byzantine-tolerant SGD algorithms based on robust statistics such as the trimmed mean. However, it is argued in [18] that these previous approaches are not specially designed for gradient descent algorithms, which results in the potential vulnerability to several specific types of attacks. To resolve the potential issues of the previous approaches based on robust statistics, the authors of [19] presented score-based approaches, which validated the updates with a standalone validation dataset, but they used a definition of Byzantine tolerance similar to that in [21]. There are also other approaches to Byzantine-tolerant SGD. For example, DRACO [23] uses redundant workers as pivots to distinguish Byzantine workers, which provides strong guarantees for Byzantine tolerance, at the cost of additional computation resources for the redundant workers. To make things worse, adding redundant workers is infeasible in federated learning scenarios. While there are many different kinds of differential privacy (DP) [24–35], we focus on local differential privacy when releasing the individual updates from the workers to the servers, which is more important in the federated learning scenarios.

## 3. Preliminaries

In this paper, we focus on distributed SGD with a parameter aerver (PS) architecture, and unreliable workers. In this section, we formally introduce the optimization problem, distributed SGD, PS architecture, and the threat model of Byzantine failures.

### 3.1. Notations

First, in Table 1, we define some important notations and terminologies that are used throughout this paper.

**Table 1.** Notations and terminologies.

| Notation/Term | Description |
| --- | --- |
| $n$ | The number of workers |
| $k$ | The number of active workers |
| $T, t$ | The number of iterations or server steps, and the current iteration $t \in [T]$ |
| $t'$ | Some previous iteration or server step, $t' \leq t - 1$ |
| $\tau$ | Maximum delay, $t - 1 - t' \leq \tau$ |
| $H$ | The number of local iterations (worker steps), or synchronization interval |
| $h$ | The current local iteration (worker step), $h \in [H]$ |
| $d$ | The number of model parameters, or the size of the model |
| $[n]$ | The set of integers $\{1, \ldots, n\}$ |
| $S_t$ | The set of randomly selected devices in the $t^{\text{th}}$ iteration |
| $b$ | Parameter of the trimmed mean |
| $H_{min}, H_{max}$ | Minimal/maximal number of local iterations |
| $H_{i,t}$ | The number of local iterations in the $t$th epoch on the $i$th device |
| $x_t$ | The initial model in the $t$th iteration |
| $x_{i,t,h}$ | Model updated in the $t$th server step and the $h$th worker step, on the $i$th device |
| $\mathcal{D}_i$ | The training dataset on the $i$th device |
| $\mathcal{D}$ | The entire training dataset $\mathcal{D} = \mathcal{D}_1 \cup \cdots \cup \mathcal{D}_n$ |
| $\mathcal{D}_r$ | The validation dataset on the validators |
| $z_{i,t,h}$ | Data (mini-batch) sampled in the $t$th server step and the $h$th worker step on the $i$th device |
| $\sigma^2$ | The variance of the stochastic gradient of a single sample |
| $f(x; z \sim \mathcal{D})$, $\nabla f(x; z \sim \mathcal{D})$ | The stochastic function value and gradient on random sample $z$, drawn from the dataset $\mathcal{D}$ (sometimes we use $f(x)$ and $\nabla f(x)$ for short) |
| $F(x), \nabla F(x)$ | $F(x) = \mathbb{E}[f(x)], \nabla F(x) = \mathbb{E}[\nabla f(x)]$, and the expectation is taken over the random samples |
| $\eta$ | Learning rate |
| $\alpha, \rho, \gamma$, etc. | Some positive constants or hyperparameters |
| $\delta$ | The approximation factor of the compressor $\mathcal{C}$, where $\|v - \mathcal{C}(v)\|^2 \leq (1 - \delta)\|v\|^2$ |
| $\|\cdot\|$ | All the norms in this paper are $l_2$-norms |
| $V_1, V_2, V_3$, etc. | Some constants defined in assumptions and used in theoretical analysis |
| $q, m$ | $q$ is the number of Byzantine workers, $m = n + q$ |
| Device | Where the training data are placed |
| Worker | The process that trains the model on the local datasets |
| Byzantine worker | Worker with Byzantine failures |
| Server | The process that maintains the global model parameters and exchanges information with the workers and validators |
| Validator | The process that validates the updates sent from the worker and filters out the potentially malicious ones |

### 3.2. Problem Formulation

We consider the following optimization problem with $n$ workers:

$$\min_{x \in \mathbb{R}^d} F(x), \tag{1}$$

where $F(x) = \frac{1}{n} \sum_{i \in [n]} F_i(x) = \frac{1}{n} \sum_{i \in [n]} \mathbb{E}_{z_i \sim \mathcal{D}_i} f(x; z_i), \forall i \in [n]$, $x \in \mathbb{R}^d$ are the set of model parameters, and $z_i$ is a mini-batch of data sampled from the local data $\mathcal{D}_i$ on the $i$th device. Each device can be a GPU, a machine with multiple GPUs, or an edge device such as a smart phone, depending on the scenario and application.

### 3.3. Distributed SGD

In this paper, we use distributed SGD to solve the optimization problem (1). In each iteration, a random mini-batch of data $z_i$ is sampled from the training dataset of any worker $i$, which is used to compute the local stochastic gradient $g_i$. We then rescale $g_i$ with the learning rate $\eta$ and update the model parameters $x$.

There are typically two strategies to execute distributed training with SGD: synchronous and asynchronous. In synchronous training, the combination of the updates aggregated from all workers are applied to the global model parameters in every step. In contrast, for asynchronous training, the global model parameters are immediately updated by any single worker without waiting for the other workers [36–38]. Typically, synchronous training is more stable with less noise, but it is also slower due to the global barrier across all workers. Asynchronous training is faster, but any asynchronous training technique needs to address instability and noisiness due to staleness.

The detailed distributed synchronous SGD algorithm is shown in Algorithm 1. In each iteration, every worker computes the stochastic gradient on a random mini-batch of data and then takes the average over the gradients from all workers. The averaged gradient is used to update the global model parameters in the same way as vanilla SGD. The global averaging incurs communication overhead.

---

**Algorithm 1** Distributed synchronous stochastic gradient descent (DS-SGD).

---

1: Initialize $x_0 \in \mathbb{R}^d$
2: **for all** iteration $t \in [T]$ **do**
3:      **for all** workers $i \in [n]$ in parallel **do**
4:          $g_{i,t} \leftarrow \nabla f(x_{t-1}; z_{i,t})$
5:      **end for**
6:      Synchronization: $\bar{g}_t \leftarrow \frac{1}{n} \sum_{j \in [n]} g_{j,t}$
7:      $x_t \leftarrow x_{t-1} - \eta \bar{g}_t$
8: **end for**

---

The detailed distributed asynchronous SGD algorithm is shown in Algorithm 2. In each iteration, the global model parameters are updated by the stochastic gradient from an arbitrary worker. Note that such a stochastic gradient is based on the model parameters from any previous iteration instead of the last iteration. There is a central node that maintains the latest version of the global model parameters. Pushing the stochastic gradient from any worker to the central node and pulling the model parameters from the central node to any worker incur communication overhead.

In brief, if the same number of stochastic gradients are applied to the global model parameters, then synchronous SGD and asynchronous SGD have the same communication overhead. The main difference is that the global updates are blocked until all gradients are collected for synchronous training, while for asynchronous training, the global updates are executed whenever the stochastic gradients arrive. Furthermore, since the stochastic gradients are potentially based on the model parameters previous to the latest version, such staleness incurs additional noise to the convergence.

---

**Algorithm 2** Distributed asynchronous stochastic gradient descent (DA-SGD).

---

1: Initialize $x_0 \in \mathbb{R}^d$
2: **for all** iteration $t \in [T]$ **do**
3:     arbitrary worker $i \in [n]$:
4:     $g_t \leftarrow \nabla f(x_{t'}; z_{i,t'}), t' < t$
5:     $x_t \leftarrow x_{t-1} - \eta g_t$
6: **end for**

---

### 3.4. Parameter-Server Architecture

For distributed SGD, there are various strategies and infrastructures that support the communication and synchronization between the workers. In this research, we focus on the parameter-server (PS) architecture [39–43], which is one of the most popular strategies to enable distributed training.

The system is composed of the server nodes and the worker nodes, as illustrated in Figure 1. Typically, the training data and the major workload of computation are distributed onto the worker nodes. For cloud computing, the worker nodes are placed on the cloud, where more worker nodes accelerate the training. For edge computing, the worker nodes are placed on the edge devices, where more worker nodes bring more training data. The server nodes, located on the cloud, are used for synchronization among the worker nodes. In summary, the workers conduct computation on their local data, and the resulting updates are then merged by the server. Such merge/synchronization operations cause the communication overhead. Different algorithms send different types of updates to the server, e.g., gradients or updated model parameters. Thus, the same PS architecture could be used in different algorithms and scenarios.

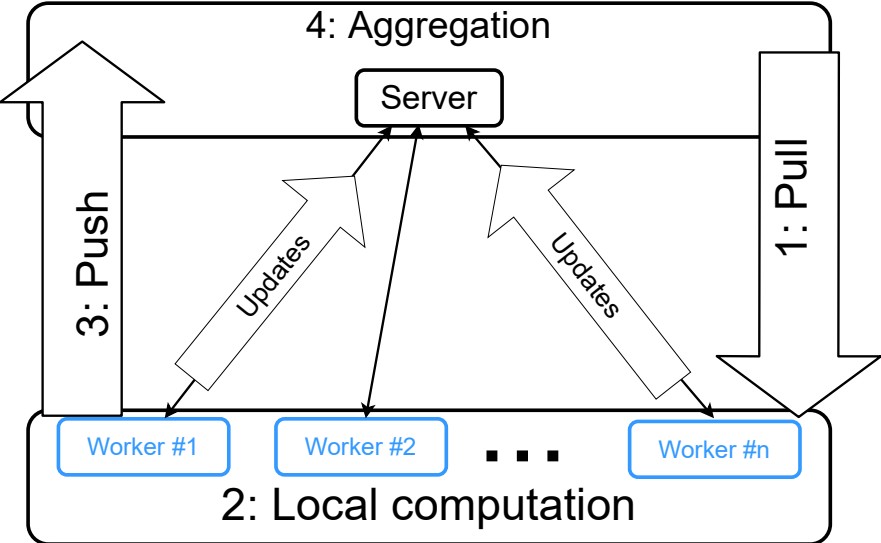

**Figure 1.** Parameter-server architecture.

### 3.5. Byzantine Failures

Byzantine failure, first introduced in [22], is a well-studied problem in the field of distributed systems. In general, Byzantine failures assume a threat model where the failed agents behave arbitrarily in a traditional distributed system. Such a threat model assumes the worse cases of failures and attacks in the distributed systems.

Many failures and attacks can be viewed as special cases of Byzantine failures. For example, the authors of [44] describe the vulnerability to bit-flipping attacks in the wireless transmission technology, where the servers can receive data via such vulnerable communication media, even if the messages are encrypted. As a result, an arbitrary fraction of the received values are corrupted. Furthermore, federated learning can be vulnerable to

data poisoning attacks, where the users feed compromised or fabricated data such as fake reviews [45] to the learning systems. A recent work [46] argues that data poisoning attacks are equivalent to Byzantine failures under certain assumptions.

The authors of [47] introduced Byzantine failures to distributed synchronous SGD, with some modifications to make the threat model fit the machine-learning scenario, in which the Byzantine failures only happen on the worker nodes. In summary, workers with Byzantine failures send arbitrary messages to other processes. We define the threat model as follows.

**Definition 1** (Threat Model). *Additional to the n honest workers, there are q Byzantine workers that send out arbitrary messages during communication. Furthermore, we assume that the workers are anonymous to the other processes.*

The existence of Byzantine workers prevents the distributed SGD from converging or decreasing the loss value on the training data. We formally define the Byzantine failures for synchronous training and asynchronous training as follows.

**Definition 2** (Threat model for synchronous SGD). *We assume that there are n honest workers and q Byzantine workers. When the server receives a gradient estimator $\tilde{g}_{i,t}$ from the ith worker in the tth iteration, it is either correct or Byzantine. If sent by a Byzantine worker, $\tilde{g}_{i,t}$ is assigned arbitrary values. If sent by an honest worker, the correct gradient is $\nabla f(x_{t-1}; z_{i,t}), \forall i \in [n], t \in [T]$. Thus, we have*

$$\tilde{g}_{i,t} = \begin{cases} arbitrary\ value, & if\ worker\ i\ is\ Byzantine,\ i.e.,\ i \in [n+1, n+q], \\ g_{i,t} = \nabla f(x_{t-1}; z_{i,t} \sim \mathcal{D}_i), & otherwise,\ i.e.,\ i \in [n]. \end{cases}$$

**Definition 3** (Threat model for asynchronous SGD). *We assume that there are n honest workers and q Byzantine workers. When the server receives a gradient estimator $\tilde{g}_t$ from the ith worker in the tth iteration, it is either correct or Byzantine. If sent by a Byzantine worker, $\tilde{g}_t$ is assigned arbitrary values. If sent by an honest worker, the correct gradient is $\nabla f(x_{t'}; z_{i,t'}), \forall i \in [n], t' \leq t - 1$. Thus, we have*

$$\tilde{g}_t = \begin{cases} arbitrary\ value, & if\ worker\ i\ is\ Byzantine,\ i.e.,\ i \in [n+1, n+q], \\ g_t = \nabla f(x_{t'}; z_{i,t'} \sim \mathcal{D}_i, t' \leq t - 1), & otherwise,\ i.e.,\ i \in [n]. \end{cases}$$

Based on the definition of Byzantine failures above, we formally define the general Byzantine tolerance for both synchronous and asynchronous SGD in the IID settings.

**Definition 4** (SGD Byzantine Tolerance). *Without a loss of generality, suppose that, in any iteration t on the server, the global model parameters are updated by $x_t = x_{t-1} - \eta u_t$, where $u_t$ is the update vector (gradient estimator) produced by different approaches. An algorithm is said to be SGD-Byzantine-tolerant if the following condition is satisfied where there are Byzantine workers in the IID settings:*

$$\exists t' \in \mathcal{T}, s.t. \langle \nabla F(x_{t'}), \quad \mathbb{E}[u_t] \rangle \geq 0,$$

*where $\mathcal{T} = \{t - 1\}$ for synchronous training, and $\mathcal{T} = \{t' : t' \leq t - 1\}$ for asynchronous training.*

In brief, an SGD-Byzantine-tolerant algorithm must have a positive inner product with the correct gradient. Note that SGD-Byzantine tolerance is a necessary condition of SGD convergence under Byzantine attacks. To guarantee the convergence sufficiently, other conditions such as smoothness, bounded variance, and $\ell_2$-norm of gradients, as well as sufficiently small learning rates are required, as we have shown in Section 5.3.

*3.6. Design Objectives*

The goal of this research is to design an integrated system resolving the critical problems in distributed machine learning: heavy communication overhead, system security, and client privacy. To be more specific, we solve the distributed optimization problem defined in (1) based on the parameter server architecture with the following additional features.

- **Communication reduction:** Optionally, the workers can reduce the communication overhead via both infrequent synchronization and message compression. To be more specific, every worker sends updates to the central server after every $H$ local iterations, and compresses the update with the error reset technique and an arbitrary compressor $\mathcal{C}$ that satisfies $\delta$-approximation: $\|\mathcal{C}(v) - v\|^2 \leq (1 - \delta)\|v\|^2, \forall v \in \mathbb{R}^d$, where $v$ is the message vector sent to the server.
- **Synchronization mode:** The system supports both synchronous and asynchronous training.
- **Byzantine tolerance:** The system supports Byzantine tolerance on the server side, with multiple choices of defense methods, including a coordinate-wise trimmed mean and score-based validation approaches.
- **Local differential privacy:** We show the theoretical guarantees that the local differential privacy of releasing updates from the workers to the servers can be achieved by randomly replacing the correct values with arbitrary (Byzantine) values. To be more specific, we use the following definition for local differential privacy (LDP).

**Definition 5** ($\xi$-LDP [48]). *Given the domain $\mathcal{D}$ of the datasets and a query function query* : $\mathcal{D} \to \mathbb{R}^d$, *a mechanism $\mathcal{M}$ with domain $\mathbb{R}^d$ is $\xi$-LDP if, for any $\mathcal{S} \subseteq Range(\mathcal{M})$ and two arbitrary datasets $D_1, D_2 \in \mathcal{D}$,*

$$\Pr[\mathcal{M}(query(D_1)) \in \mathcal{S}] \leq \exp(\xi) \Pr[\mathcal{M}(query(D_2)) \in \mathcal{S}].$$

In the case of distributed SGD, the queries are the gradients or updates released by the workers. Note here that the result does not refer to differential privacy of the full training pipeline, but only to differential privacy of a single step with the following threat model: We assume that the attackers are the curious servers, from which we want to protect the workers and of which we want to avoid the servers recovering the private updates from individual workers. There are various mechanisms or protocols that achieve LDP [49–56]. Furthermore, some LDP mechanisms can also provide $\xi$-DP guarantees [48,57] for the full training process, at the expense of increased $\xi$ for LDP.

## 4. Methodology

In this section, we present ZenoPS, which is a distributed learning system based on the PS architecture. Our implementation is based on the cutting-edge PS implementation called BytePS [42,43]. In the ZenoPS architecture, the processes are categorized into three roles: servers, workers, and validators. Note that, compared to the original PS architecture, ZenoPS has an additional role of nodes: **validators**, which read the update cached on the servers and filter out the potentially malicious ones. The responsibilities of the three roles are described as follows:

- **Server:** The server nodes maintain the global model parameters. ZenoPS also supports multiple server nodes, where the model parameters are partitioned into several blocks and are uniquely assigned to different server nodes via some hash functions. The servers communicate to both the workers and the validators. On one hand, the servers send the latest model parameters to the workers on request and cache the updates sent from the workers. On the other hand, the servers send the cached updates to the validators and collect the verified updates from the validators. Once verified, the updates are supposed to be benign and safe for the servers to update the global model parameters. In the synchronous mode, the servers will wait for all validators to respond, take the average of the verified updates from all validators, and then apply

the averaged updates to the global model parameters. In the asynchronous mode, the servers update the global model parameters whenever a verified update arrives.

- **Worker:** The worker nodes take the main workload of computation in the ZenoPS system. Periodically, the workers pull the latest model parameters from the servers, run SGD locally for $H$ steps to update the local models, and then send the accumulated updates to the server. Optionally, the workers can compress the updates sent to the servers and use error reset for biased compressors. Optionally, the workers can randomly replace the updates sent to the servers with arbitrary noise, which helps provide privacy preservation. We show that the adverse effects of adding noise are managed by the robust aggregation.

- **Validator:** The validators are used for filtering out the potentially malicious updates. Any update sent from the workers will be cached and rallied to a validator. The users of ZenoPS can choose various algorithms for validation. In the synchronous mode, the validators can use the coordinate-wise trimmed mean, Phocas [17], or score-based approaches such as Zeno [19], or they can simply take the average. In the asynchronous mode, the validators have two options: score-based approaches or no validation at all (approving any received updates). Future robust training approaches can be implemented using the same framework. Typically, we assume fast communication between the servers and validators. A reasonable configuration is to co-locate the servers and the validators. For example, when using the score-based approaches, a validator node should have the computation power similar to a worker node. In that case, we could put the server node and the validator node in the same machine, where the server is assigned to the CPU, and the validator is assigned to the GPU.

The detailed algorithm is shown in Algorithm 3, where we only use the score-based validation approach defined in Definition 6 for Byzantine tolerance.

The workers pull the model parameters from the servers and add the local residual errors to the pulled model. After $H$ steps of local updates, the workers obtain the accumulated local update, which is the difference between the current version of local models and the previously pulled models. The workers will then compress the local updates, update the local residual errors, and send the compressed updates to the servers. To protect privacy, the workers can randomly replace the messages with arbitrary values, which inserts some noise in the released data, in order to achieve differential privacy.

The servers simply relay all updates sent from the workers to the validators and respond to any pulling request from the workers and the validators. In the synchronous mode, the servers will wait until the approved updates are received from all validators. In the asynchronous mode, the servers will update the global model parameters whenever an approved update is received from any validator. Note that the validators can send vectors of all 0 values to the server, as a notification that an update fails the validation. The servers will not move on to the next iteration until a non-zero approved update is received.

The validator uses the criterion defined in Definition 6 to validate the candidate updates sent from the servers. If a candidate update passes the validation procedure, it will be sent to the servers; otherwise the validator will send a vector of all 0 values to the server. Periodically, the validators will pull the model parameters from the servers and update the validator vector. Note that, in the synchronous mode, the validators always pull the latest version of the global model; in the asynchronous mode, the pulled model can have some delay.

---

**Algorithm 3** ZenoPS with score-based validation.

---

1: **Server**
2: Initialize $x_0 \in \mathbb{R}^d$
3: **for all** server step $t \in [T]$ **do**
4:     $u_t \leftarrow 0$
5:     **while** $u_t = 0$ **do**
6:         Send $x_{t-1}$ to workers on request
7:         Receive updates $\{\tilde{u}_i : i \in [n+q]\}$ from workers, and relay to validators
8:         **if** Synchronous **then**
9:             $u_t \leftarrow \frac{1}{n_v} \sum_{i \in [n_v]} u_i$, after all validators respond
10:       **else**
11:          $u_t \leftarrow u_i$, after receiving $u_i : i \in [n_v]$ from a validator
12:       **end if**
13:       Update the parameters $x_t \leftarrow x_{t-1} + \alpha u_t$
14:     **end while**
15: **end for**

---

1: **Worker (honest)** $i = 1, \ldots, n$
2: Initialize $e_i \leftarrow 0$
3: **while** until terminated **do**
4:     Receive $x_{t'}$ from the server, and initialize $x_{i,t',0} \leftarrow x_{t'} + e_i$
5:     **for all** local iteration $h \in [H]$ **do**
6:         $x_{i,t',h} \leftarrow x_{i,t',h-1} - \eta \nabla f(x_{i,t',h-1}; z_i \sim \mathcal{D}_i)$
7:     **end for**
8:     Compute the accumulated update $u_i \leftarrow x_{i,t',H} - x_{t'}$
9:     Compress $u_i' \leftarrow \mathcal{C}(u_i)$, and update the local residual error $e_i \leftarrow u_i - u_i'$
10:     Replace $u_i'$ by arbitrary value with probability $p_{byz}$ (optional)
11:     Send $u_i'$ to the server
12: **end while**

---

1: **Validator** $j = 1, \ldots, n_v$
2: Initialize $counter = 1$
3: **while** until terminated **do**
4:   **if** $\mathrm{mod}\,(counter, n/n_v) = 1$ **then**
5:     Pull $x_{t'}$ from server, update for $H$ SGD steps, and obtain $x_{t',H}$
6:     Update the validation vector $v_t = x_{t',H} - x_{t'}$
7:   **end if**
8:   $counter \leftarrow counter + 1$
9:   Wait until any update $\tilde{u}$ arrives
10:   $u \leftarrow \tilde{u}$ if the validation defined in Definition 6 is passed, else $u \leftarrow 0$
11:   Send $u$ to the server
12: **end while**

---

To make the system design and the relationship between the three groups of nodes clearer, we illustrate the ZenoPS architecture in Figure 2.

In the remainder of this section, we present more details of the optional features provided by ZenoPS.

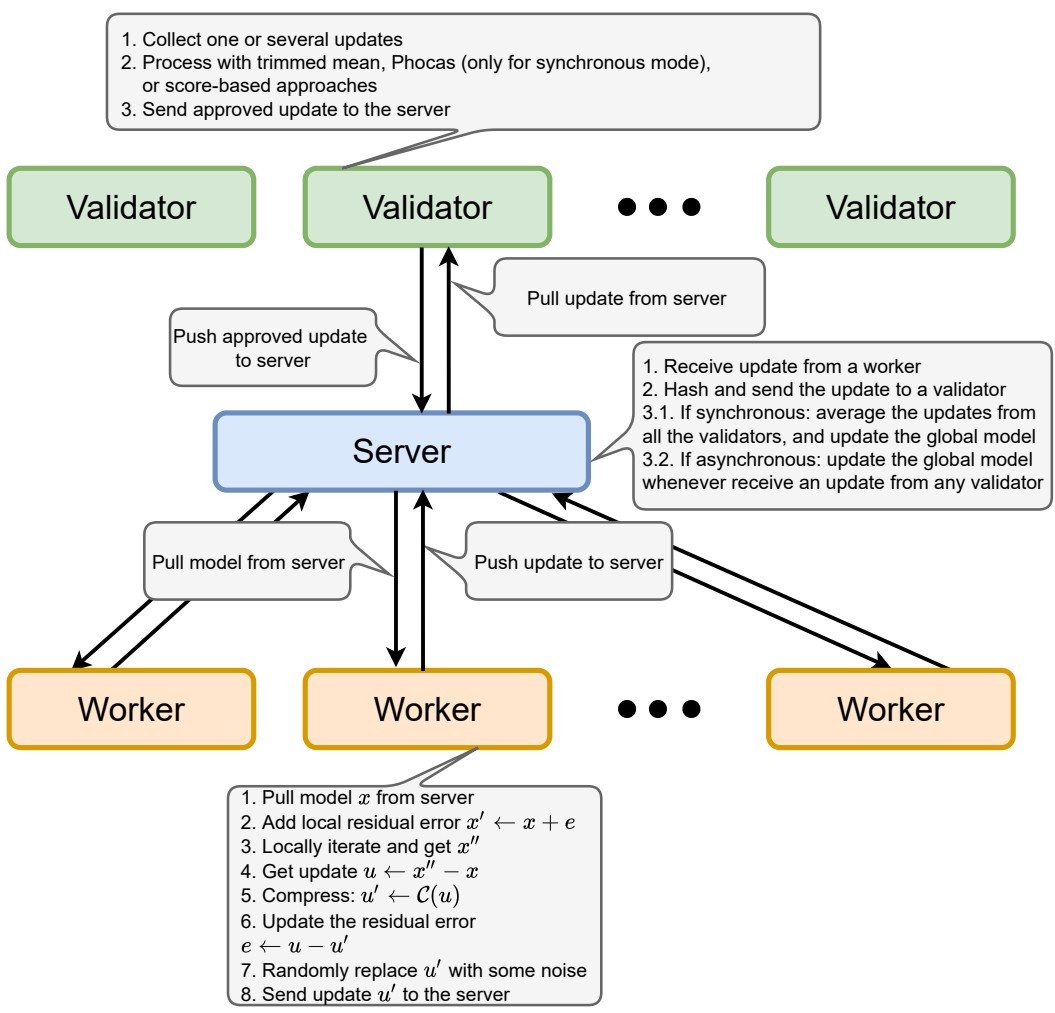

**Figure 2.** ZenoPS architecture.

*4.1. Communication Reduction*

When sending updates to the servers, the workers can choose to compress the message using an arbitrary compressor $\mathcal{C}$ and use the technique of error reset to handle the residual error of the compression. In the algorithm, we compress the locally accumulated update $u_i$ and maintain the local residual error $e_i$ of worker $i$, in Line 9 in Algorithm 3 (in the worker process). The local residual error will then be applied to the model parameters pulled from the servers in the next time. Similar to CSER in [15], we assume $\delta$-approximate compressors: $\|\mathcal{C}(v) - v\|^2 \leq (1 - \delta)\|v\|^2, \forall v \in \mathbb{R}^d$, where $\delta \in [0, 1]$. The choice of the expected compression ratio $\delta$ depends on the sensitivity of the deep neural network and the networking environment. When the number of workers increases, the network tends to become congested and requires a larger compression ratio to maintain the efficiency of training. However, larger compression ratios usually incur larger noise. Thus, case-by-case hyperparameter tuning is typically required to find the optimal trade-offs.

*4.2. Synchronization Modes*

Besides the synchronous mode, ZenoPS also provides the asynchronous mode, where the workers and validators can check in and send updates to the server at any time. The servers launch multiple threads to handle the pushing and pulling request sent by the workers and validators. In the synchronous mode, any pulling request from the workers or the validators will be blocked until the global model parameters are updated. In the

asynchronous mode, the servers respond to the pulling requests immediately without any barrier.

If the workers have almost identical computation power and networking environments, the synchronous mode is recommended, which is more efficient and stable. However, there are also cases where the workers have heterogeneous capabilities and communication speeds, especially in the federated learning scenario. The heterogeneity results in stragglers in the system, which then causes noisy or slow updates sent to the servers. In that case, the asynchronous mode is recommended for efficiency. The noise caused by the stragglers can be mitigated by tuning the mixing hyperparameter $\alpha$ and using the validators to automatically filter out the extremely outdated updates.

### 4.3. Byzantine Tolerance

In ZenoPS, we use the following score-based approach for Byzantine tolerance.

**Definition 6** (ZenoPS validation). *Assume that, in the tth iteration, based on the model parameters $x_{t'}$ (with latency) on the server, where $t' \leq t - 1$ ($t' = t - 1$ in the synchronous mode), the validator locally updates the model parameters for H steps using the validation data and obtains the accumulated update for validation: $v = -\eta \sum_{h \in H} \nabla f(x_{t',h-1}; z_{t',h} \sim \mathcal{D}_r)$, where $x_{t',h} = x_{t',h-1} - \eta \nabla f(x_{t',h-1}; z_{t',h} \sim \mathcal{D}_r)$, and $x_{t',0} = x_{t'}$. An update u from any worker will be approved and sent back to the server for updating the global model parameters if the following two conditions are satisfied:*

$$\langle u, v \rangle \geq \rho \|v\|^2 + \epsilon,$$
$$\|u\|^2 \leq (1 + \gamma)\|v\|^2 \quad \text{alternative: clip } u \text{ to the maximum norm } (1 + \gamma)\|v\|^2 ,$$

*where $\rho, \gamma > 0$ are some hyperparameters for the thresholds.*

The above validation mechanism allows both the candidate update and the validation vector to be the accumulation of multiple steps of SGD updates. Thus, such a validation mechanism can be used for distributed SGD with infrequent synchronization, such as federated optimization or local SGD.

### 4.4. Byzantine Mechanism and Privacy Preservation

In our implementation, we add a simulator on the worker side to simulate Byzantine failures and send malicious values to the servers. It turns out that such a simulator can also be used to generate random noises for privacy preservation. On the server side, the random noises will then be treated as values from Byzantine workers. Formally, we introduce the Byzantine mechanism as follows.

**Definition 7.** *In the Byzantine mechanism denoted as $\mathcal{M}_{byz}$, the ith worker sends the vector*

$$\tilde{v}_i = \mathcal{M}_{byz}(v_i) = \begin{cases} v_i & \text{with probability } 1 - p_{byz}; \\ \text{noise} \sim \mathcal{D}_{noise} & \text{otherwise,} \end{cases}$$

*where $v_i$ is the correct vector. In other words, with probability $p_{byz}$, the worker sends the value noise, which is randomly drawn from the distribution $\mathcal{D}_{noise}$ instead of the correct value $v_i$, to the server.*

## 5. Theoretical Analysis

In this section, we show the theoretical guarantees of ZenoPS using the score-based approach in the validators. We theoretically analyze the Byzantine tolerance, the convergence, and the differential privacy of the proposed system. The detailed proofs of the theorems are in Appendix A.

### 5.1. Assumptions

**Assumption 1** (Smoothness). *$F(x)$ and $F_i(x), \forall i \in [n]$ are L-smooth:*

$$F(y) - F(x) \leq \langle \nabla F(x), y - x \rangle + \frac{L}{2} \|y - x\|^2, \forall x, y,$$

*and*

$$F_i(y) - F_i(x) \leq \langle \nabla F_i(x), y - x \rangle + \frac{L}{2} \|y - x\|^2, \forall x, y.$$

**Assumption 2** (Variance). *For any stochastic gradient $g_i = \nabla f(x; z_i), z_i \sim \mathcal{D}_i$, we assume bounded intra-worker variance: $\mathbb{E}[\|g_i - \nabla F_i(x)\|^2] \leq V_1 = \frac{\sigma^2}{s}, \forall x \in \mathbb{R}^d$, where $\sigma^2$ is the variance of the gradient of a single data sample, and s is the mini-batch size per worker.*

**Assumption 3** (Bounded gradients). *For any stochastic gradient $\nabla f(x; z), \forall x \in \mathbb{R}^d$, we assume bounded expectation: $\|\nabla F(x; z)\|^2 = \|\mathbb{E}[\nabla f(x; z)]\|^2 \leq V_1', \forall i \in [n], t \in [T]$. In some cases, we directly assume the upper bound of the stochastic gradients: $\|\nabla f(x; z)\|^2 \leq V_3$.*

**Assumption 4** (Global minimum). *There is at least one global minimum $x_*$, where $F(x_*) \leq F(x), \forall x$.*

### 5.2. Byzantine Tolerance

In the following theorem, we show the Byzantine tolerance of score-based validation approach defined in Definition 6.

**Theorem 1** (SGD-Byzantine tolerance of ZenoPS). *Assume that the validation data is close to the training data $\|\nabla F_r(x) - \nabla F(x)\| \leq r, \forall x \in \mathbb{R}^d$ where $r \geq 0$ and $F_r(x) = \mathbb{E}[f_r(x; z \sim \mathcal{D}_r)]$ (the expectation is taken with respect to z), and $\epsilon$ is large enough. Under Assumptions 2 and 3 (bounded gradients and variance, which is also applied to the validation gradients), for a verified update u sent from the validators, there is a $t' \leq t - 1$ such that we have the positive inner-product:*

$$\left\langle -\eta \sum_{h \in H} \nabla F(x_{t', h-1}), \mathbb{E}[u] \right\rangle \geq 0,$$

*where $x_{t', h} = x_{t', h-1} - \eta \nabla f(x_{t', h-1}; z_{t', h} \sim \mathcal{D}_r)$, and $x_{t', 0} = x_{t'}$.*

### 5.3. Convergence

Now we prove the convergence of the proposed algorithm in the synchronous mode. For simplicity, we take $\alpha = 1$ in the synchronous mode.

**Theorem 2** (Error bound of ZenoPS in the synchronous mode without compression). *In addition to Assumption 1 (smoothness), Assumption 2 (bounded variance), Assumption 3 (bounded gradients), and Assumption 4 (global minimum), we assume that the compressors are disabled and that the validation data is close to the training data $\|\nabla F_r(x) - \nabla F(x)\| \leq r, \forall x \in \mathbb{R}^d$ where $r \geq 0$ and $F_r(x) = \mathbb{E}[f_r(x; z \sim \mathcal{D}_r)]$ (the expectation is taken with respect to z). Taking $\epsilon = cH^2\eta^2$, we have the following error bound for ZenoPS in the synchronous mode:*

$$\frac{\sum_{t \in [T]} \mathbb{E}\|\nabla F(x_{t-1})\|^2}{T}$$

$$\leq \frac{F(x_0) - F(x_*)}{TH\eta\rho} + \frac{3 + 3\gamma + 4\rho}{2\rho} LH\eta V_3 + \frac{(4 + 3\gamma + 4\rho)r\sqrt{V_3} + r^2 - 2c}{2\rho} + \frac{(1 + \gamma)\sqrt{V_3}\sigma}{\rho\sqrt{s_r}}.$$

*Taking $\eta = \frac{1}{\sqrt{TH}}$, $s_r \propto TH$, we have*

$$\frac{\sum_{t \in [T]} \mathbb{E}\|\nabla F(x_{t-1})\|^2}{T}$$

$$\leq \mathcal{O}\left(\frac{F(x_0) - F(x_*)}{\sqrt{TH}\rho}\right) + \mathcal{O}\left(\frac{H}{\sqrt{TH}\rho}\right) + \mathcal{O}\left(\frac{\sigma}{\sqrt{TH}\rho}\right) + \frac{(4 + 3\gamma + 4\rho)r\sqrt{V_3} + r^2 - 2c}{2\rho}.$$

The error bound of ZenoPS is composed of four parts: the gap to the initial value $[F(x_0) - F(x_*)]$, the error caused and the infrequent synchronization that is proportional to $H$, the noise caused by variance $\sigma$, and the validation error caused by Byzantine tolerance $\frac{(4+3\gamma+4\rho)r\sqrt{V_3}+r^2-2c}{2\rho}$. A better validation dataset that is closer to the training dataset with a smaller $r$ decreases the validation error. Increasing $c$ and $\rho$ or decreasing $\gamma$ decreases the validation error, but also potentially decreases the chances that benign updates pass the validation procedure, which can slow down the optimization progress. In short, stronger security guarantees smaller validation error, but also slower convergence.

When the compressors are enabled, there is an additional error term with the approximation factor $\delta$.

**Theorem 3** (Error bound of ZenoPS in the synchronous mode with compression). *In addition to Assumption 1 (smoothness), Assumption 3 (bounded gradients), and Assumption 4 (global minimum), we assume that the compressors are disabled and the validation data is close to the training data $\|\nabla F_r(x) - \nabla F(x)\| \leq r, \forall x \in \mathbb{R}^d$ where $r \geq 0$ and $F_r(x) = \mathbb{E}[f_r(x; z \sim \mathcal{D}_r)]$ (the expectation is taken with respect to z). Taking $\epsilon = cH^2\eta^2$ and $\eta = \frac{1}{\sqrt{TH}}$, we have the following error bound for ZenoPS in the synchronous mode:*

$$\frac{\sum_{t \in [T]} \mathbb{E}\|\nabla F(x_{t-1})\|^2}{T}$$

$$\leq \mathcal{O}\left(\frac{F(x_0) - F(x_*)}{\sqrt{TH}\rho}\right) + \mathcal{O}\left(\frac{H}{\sqrt{TH}\rho}\right) + \mathcal{O}\left(\frac{\sigma}{\sqrt{TH}\rho}\right) + \frac{(4 + 3\gamma + 4\rho)r\sqrt{V_3} + r^2 - 2c}{2\rho}$$

$$+ \frac{(1 - \delta)(1 + \gamma)L\sqrt{H}V_3}{2\rho\sqrt{T}\left(1 - \sqrt{1 - \delta}\right)^2}.$$

Finally, we prove the convergence of the proposed algorithm in the asynchronous mode.

**Theorem 4** (Error bound of ZenoPS in the asynchronous model with compression). *In addition to Assumption 1 (smoothness), Assumption 2 (bounded variance), Assumption 3 (bounded gradients), and Assumption 4 (global minimum), we assume that the validation data is close to the training data $\|\nabla F_r(x) - \nabla F(x)\| \leq r, \forall x \in \mathbb{R}^d$ where $r \geq 0$ and $F_r(x) = \mathbb{E}[f_r(x; z \sim \mathcal{D}_r)]$ (the expectation is taken with respect to z). Furthermore, we assume that, in any server step t, the approved update is based on the global model parameters in the server step $t'$, where $t' \leq t - 1$ has bounded delay $t - 1 - t' \leq \tau$. Taking $\epsilon = cH^2\eta^2$, we have the following error bound for ZenoPS in the asynchronous mode:*

$$\frac{\sum_{t \in [T]} \mathbb{E}\|\nabla F(x_{t-1})\|^2}{T}$$

$$\leq \frac{F(x_0) - F(x_*)}{TH\alpha\eta\rho} + \frac{(2\tau + 2 + \alpha)(1 + \gamma) + 4(\tau + 1)\rho}{\rho}LH\eta V_3$$

$$+ \frac{(2 + \gamma + 4\rho)r\sqrt{V_3} + r^2 - 2c}{2\rho} + \frac{(1 + \gamma)\sqrt{V_3}\sigma}{\rho\sqrt{s_r}} + \frac{(1 - \delta)(1 + \gamma)L\sqrt{H}V_3}{2\rho\sqrt{T}\left(1 - \sqrt{1 - \delta}\right)^2}.$$

*Taking $\eta = \frac{1}{\sqrt{TH}}$, $s_r = TH$, we have*

$$\frac{\sum_{t\in[T]}\mathbb{E}\|\nabla F(x_{t-1})\|^2}{T}$$

$$\leq \frac{F(x_0)-F(x_*)}{\sqrt{TH}\alpha\rho} + \sqrt{\frac{H}{T}\frac{(2\tau+2+\alpha)(1+\gamma)+4(\tau+1)\rho}{\rho}}LV_3$$

$$+\frac{(2+\gamma+4\rho)r\sqrt{V_3}+r^2-2c}{2\rho} + \frac{(1+\gamma)\sqrt{V_3}\sigma}{\rho\sqrt{TH}} + \frac{(1-\delta)(1+\gamma)L\sqrt{H}V_3}{2\rho\sqrt{T}\left(1-\sqrt{1-\delta}\right)^2}.$$

*5.4. Local Differential Privacy*

Finally, we present the theoretical guarantee of the Byzantine mechanism in local differential privacy. Denote $p_{noise}(z)$ as the probability density function of the random variable $noise \sim \mathcal{D}_{noise}$. We show that the Byzantine mechanism is LDP in the following theorem.

**Theorem 5** (LDP of Byzantine mechanism). *Assume that the noise distribution $\mathcal{D}_{noise}$ has the support $[a,b]$, and $p_- = \min_{z\in[a,b]} p_{noise}(z) > 0$, where $p_{noise}(\cdot)$ is the PDF of the noise distribution $\mathcal{D}_{noise}$ (e.g., uniform distribution with support $[a,b]$). The Byzantine mechanism is then $\xi$-LDP, where*

$$\xi = \frac{1-p_{byz}}{p_{byz}p_-}.$$

Thus, a larger $p_{byz}$ makes the mechanism more differentially private, at the cost of replacing more correct values with the random noise as well as a slowdown of the overall optimization progress on the servers.

## 6. Experiments

In this section, we empirically evaluate the proposed ZenoPS system in various simulated settings. We test the performance of ZenoPS in both the synchronous and asynchronous mode, where the asynchronous experimental setting represents edge computing with flexible workers, and the synchronous experimental setting represents a traditional datacenter. Furthermore, we test the robustness of ZenoPS under various attacks in both synchronous and asynchronous modes.

*6.1. Evaluation Setup*

We trained ResNet-20 on the CIFAR-10 [58] dataset. The mini-batch size was 32. In this section, our experiments were conducted in real distributed environments with CPU workers. We used $\eta = 0.2$ in the synchronous mode and $\eta = 0.1$ in the asynchronous mode. In the epochs 100 and 150, the learning rate decayed by 0.1 in the synchronous mode and 0.2 in the asynchronous mode. We used a constant $\alpha = 1$ in the synchronous mode. In the asynchronous mode, we used an initial value $\alpha = 0.4$, which decayed by 0.5 in the epochs 100 and 150.

The communication overhead was reduced by both message compression and infrequent synchronization. We used random block-wise sparsification to compress the communication. Whenever a worker sent an update of a block of parameters to the server, with the probability of 0.2, it ignored the communication and put the update into the local residual error. Furthermore, the number of local steps $H$ was 8.

For the validators, we set $b = 3$ for both the trimmed mean and Phocas. We set $\gamma = 0.6$, $\rho = -0.001$, $\epsilon = 0$ for Zeno validation in the synchronous mode. In the asynchronous mode, we set $\gamma = 0$ (with clipping), $\rho = -0.02$, $\epsilon = 0$ for Zeno validation.

We randomly and evenly partitioned the training data into the number of workers plus one parts and assigned the additional part to every validator.

We evaluated ZenoPS in both the synchronous mode and the asynchronous mode. In the synchronous mode, we used `Mean`, `Trimmed mean`, `Phocas` [17], and `Zeno` (the score-

based validation defined in Definition 6) as the validators. In the asynchronous mode, we used `FedAsync` (no validation) and `Zeno` as the validators.

In the experiments, we used the following two types of attacks:

- **Sign-flipping attack:** The worker multiplies the original updates by $-\zeta$, i.e., flips the sign and rescales the updates by $\zeta$. We call this type of attack a "sign-flipping attack rescaled by $\zeta$". The same type of attacks have also been used in previous work [20,23].
- **Random attack:** The worker uses Gaussian random values with a 0 mean to replace the original values. If we use Gaussian random values with variance $\zeta$, then we call this type of attack a "random attack rescaled to $\zeta$". On the other hand, if we use Gaussian random values and rescale the Byzantine vector, so that the $\ell_2$ norm of the Byzantine vector is $\zeta$ times the original one, then we call this type of attack a "random attack rescaled by $\zeta$". The same type of attacks have also been used in previous work [47].

We simulated the Byzantine attacks by randomly replacing the vectors sent from the workers to the servers with a probability of 0.2.

### 6.2. Empirical Results

The result of ZenoPS in the synchronous mode is shown in Figure 3. We can see that, when there are no Byzantine attacks, all algorithms have similar performance. When there are Byzantine attacks, using a Zeno validator, ZenoPS converged almost as well as using averaging without any attack. However, the trimmed mean had relatively bad results under sign-flipping attacks, and both the trimmed mean and Phocas failed under the random attacks rescaled to 1. In some cases, where the attacks were relatively weak, such as the sign-flipping attacks rescaled by 6 and the random attacks rescaled by 8, Phocas also had good convergence, which is an option that is cheaper than Zeno. However, in general, using Zeno as the validator provided the best performance under all kinds of attacks.

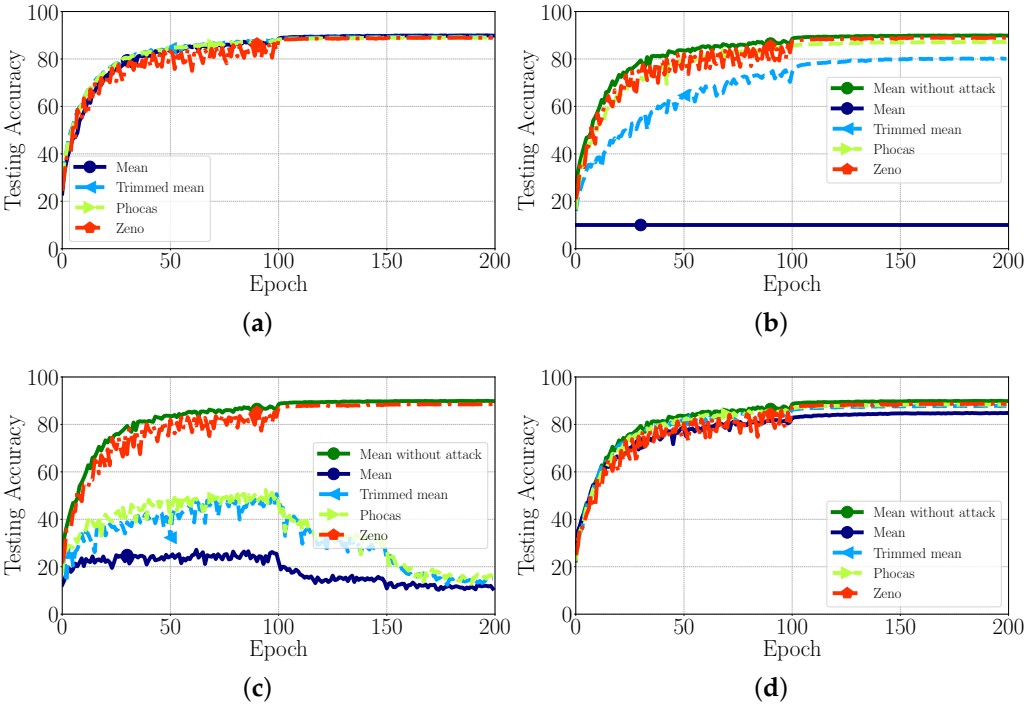

**Figure 3.** ZenoPS in the synchronous mode with various attacks and validators. In each experiment, we launched 1 server, 1 validator, and 16 workers. In each iteration, each worker had a 50% probability of being active. (**a**) No attacks. (**b**) Sign-flipping attack rescaled by 6. (**c**) Random attack rescaled to 1. (**d**) Random attack rescaled by 8.

Furthermore, we show that ZenoPS is capable of using multiple servers and validators. Different servers are responsible for different blocks of the model parameters. The updates sent to the servers will be evenly hashed to the validators, so that different validators are assigned a similar workload. Figure 4 shows that, by using multiple validators, ZenoPS has a similar performance compared to the case of using a single validator. Note that the multiple validators send updates to the server independently and asynchronously. Such asynchronicity causes additional noise compared to the case where there is only 1 Zeno validator. Furthermore, without changing the hyperparameter $b = 3$ for the trimmed mean and Phocas validators, adding more validators results in more potentially harmful updates being dropped. Hence, the trimmed mean and Phocas validators have slightly better results in Figure 4b than those in Figure 3c.

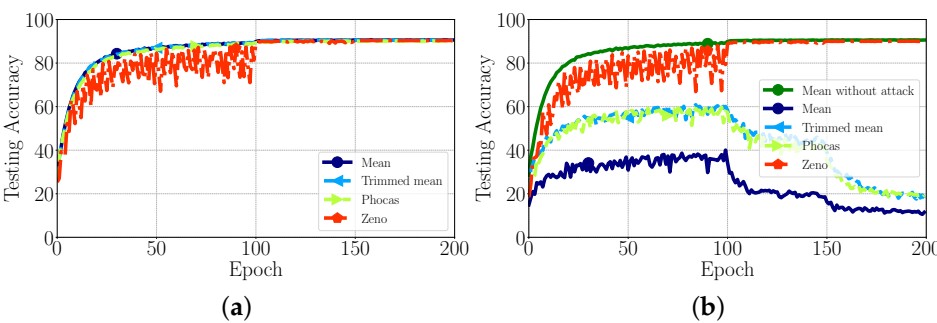

**Figure 4.** ZenoPS in the synchronous mode with multiple validators. In each experiment, we launched 2 servers, 2 validators, and 16 workers. In each iteration, all workers were active. (**a**) No attacks. (**b**) Random attack rescaled to 1.

The result of ZenoPS in the asynchronous mode is shown in Figure 5. When there are no Byzantine attacks, using Zeno as the validator provides good convergence similar to FedAsync. Adding Byzantine attacks makes FedAsync performs much worse. Using Zeno as the validator, ZenoPS converges as well as the cases where there are no attacks. We also show the results where two servers and two validators are used in Figure 6.

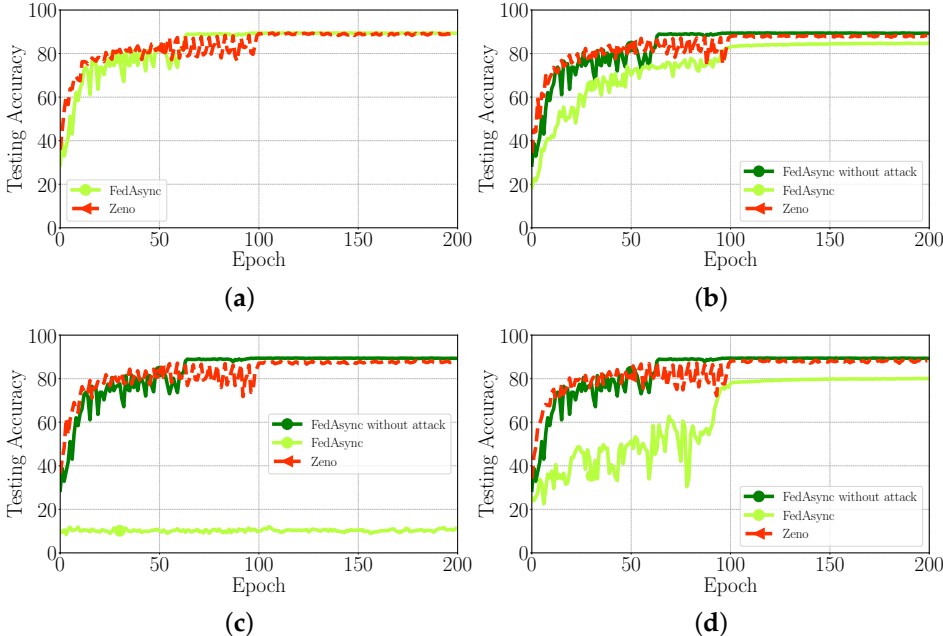

**Figure 5.** ZenoPS in the asynchronous mode with various attacks and validators. In each experiment, we launched 1 server, 1 validator, and 8 workers. In each communication round, each worker had a 50% probability of dropping the entire communication. (**a**) No attacks. (**b**) Sign-flipping attack rescaled by 2. (**c**) Random attack rescaled to 1. (**d**) Random attack rescaled by 8.

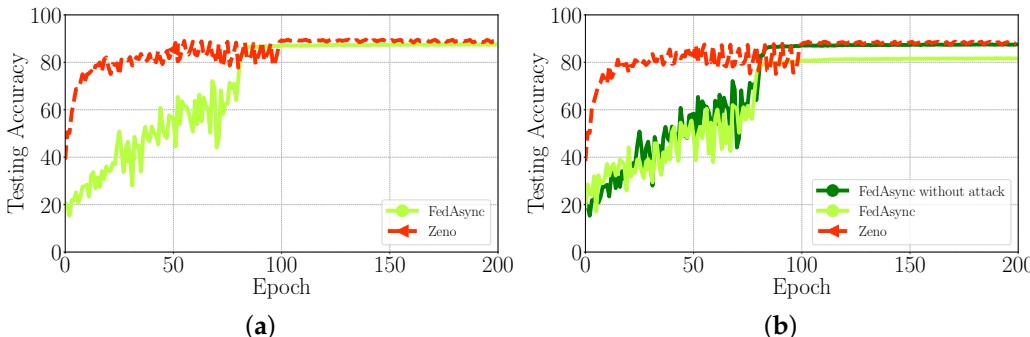

**Figure 6.** ZenoPS in the asynchronous mode with multiple validators. In each experiment, we launched 2 servers, 2 validators, and 8 workers. (**a**) No attacks. (**b**) Random attack rescaled by 8.

As shown in Figure 7, we tested the sensitivity to the hyperparameter $\rho$, with fixed $\epsilon = 0$. We varied $\rho$ in $\{0.1, -0.01, -0.001, 0, 0.001, 0.01, 0.1\}$. In most cases, the Zeno validator was insensitive to $\rho$ and converged to the same value. The only exception was the case of $\rho = 0.1$, where the threshold was too large and made the convergence extremely slow. Thus, while a grid search needs to be performed for hyperparameter tuning in practice, for $\rho$ we recommend using a negative value close to 0. Although the convergence analysis shows that a larger $\rho$ yields smaller error bounds, it will also decrease the number of approved updates. As shown in Figure 7, when $\rho = 0.1$, only 16% of the updates passed the validation. By decreasing the threshold, the approval rate approached the ideal value of 80% (20% of the updates are Byzantine). We also show the result of training the model only using the validation data, which is referred to as "validation only." Note that, in this case, the training was not affected by the attacks. It is shown that, if we only use the validation data, the testing accuracy will be very low. Thus, when using Zeno as the validator, the models learn from the training data. However, if the threshold $\rho$ is too large, the approved updates will be extremely biased to the validation data, which causes the performance to be close to the case of "validation only." Another interesting observation is that "validation only" has a slightly higher testing accuracy at the very beginning of the training. Thus, the validation dataset is useful for training the model for several epochs as initialization.

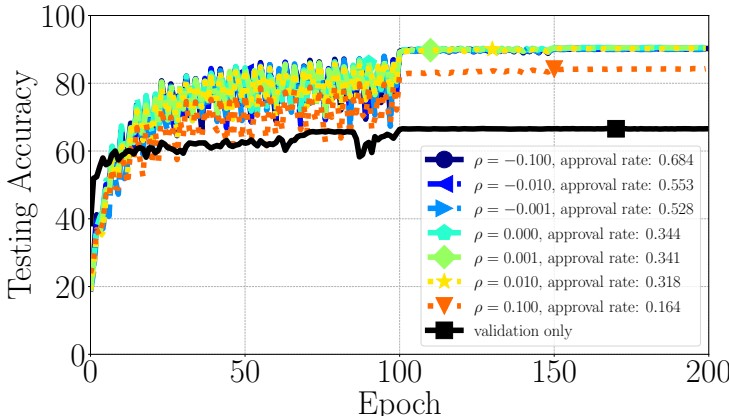

**Figure 7.** Sensitivity to $\rho$. $\rho$ varies, with fixed $\epsilon = 0$ and $\gamma = 0.6$. ZenoPS in the synchronous mode with multiple validators. In each experiment, we launched 2 servers, 2 validators, and 16 workers. Random attack is rescaled to 1.

## 7. Conclusions

We propose a prototype of a distributed learning system, ZenoPS, that integrates message compression, infrequent synchronization, both asynchronous and synchronous training, and score-based validation. The proposed system provides communication

reduction, asynchronous training, Byzantine tolerance, and local differential privacy, with theoretical guarantees, and was evaluated on an open benchmark.

This work also raises some open problems to be solved in future work. One limitation of ZenoPS is the relatively high computation overhead of the score-based validation algorithm, which gives improved protection from attacks at the cost of consuming more computation resources. How the score-based validation can be more efficient remains to be explored. The learning system could also be made to automatically and adaptively choose the compression ratio and validation methods for different training tasks, to make the system more user-friendly in practice. For theoretical analysis, it will be interesting to establish theories for optimal values of hyperparameters such as $\rho$ in score-based validation, and to study how hyperparameters such as the learning rate $\eta$ affect the optimal value of $\rho$.

**Author Contributions:** Conceptualization, C.X., O.K., and I.G.; methodology, C.X., O.K., and I.G.; software, C.X.; resources, O.K. and I.G.; writing—original draft preparation, C.X.; writing—review and editing, O.K. and I.G.; visualization, C.X.; supervision, O.K. and I.G.; project administration, O.K. and I.G.; funding acquisition, O.K. and I.G. All authors have read and agreed to the published version of the manuscript.

**Funding:** This research was funded by J.P. Morgan 2020 AI Research Ph.D. Fellowship Awards 098804, a 2021 Sloan Fellowship, and a gift from Microsoft. This work was also funded in part by NSF 2046795, 1909577, 1934986, and NIFA award 2020-67021-32799.

**Data Availability Statement:** This research uses the following publicly archived dataset: the CIFAR-10 [58] dataset, available on the website https://www.cs.toronto.edu/kriz/cifar.html (accessed on 15 February 2021).

**Acknowledgments:** We would like to thank Intel and Microsoft Azure for granting the computation resources to support this research.

**Conflicts of Interest:** The funders had no role in the design of the study, in the collection, analyses, or interpretation of data, in the writing of the manuscript, or in the decision to publish the results.

## Appendix A. Proofs

**Theorem A1** (SGD-Byzantine Tolerance of ZenoPS)**.** *Assume that the validation data is close to the training data* $\|\nabla F_r(x) - \nabla F(x)\| \leq r, \forall x \in \mathbb{R}^d$ *where* $r \geq 0$ *and* $F_r(x) = \mathbb{E}[f_r(x; z \sim \mathcal{D}_r)]$ *(the expectation is taken with respect to z), and* $\epsilon$ *is large enough. Under Assumptions 2 and 3 (bounded gradients and variance, which is also applied to the validation gradients), for a verified update u sent from the validators, there is a* $t' \leq t - 1$ *such that we have the positive inner-product:*

$$\left\langle -\eta \sum_{h \in H} \nabla F(x_{t',h-1}), \mathbb{E}[u] \right\rangle \geq 0,$$

*where* $x_{t',h} = x_{t',h-1} - \eta \nabla f(x_{t',h-1}; z_{t',h} \sim \mathcal{D}_r)$, *and* $x_{t',0} = x_{t'}$.

**Proof.** Using $\|\nabla F_r(x) - \nabla F(x)\| \leq r$, it is easy to check that $\langle \nabla F_r(x), \nabla F(x) \rangle \in [-\frac{r^2}{2}, \frac{r^2}{2}]$. The algorithm guarantees that there is a $t' \leq t - 1$ such that $\langle -\eta \sum_{h \in H} \nabla f_r(x_{t',h-1}), u \rangle \geq \rho \| \eta \sum_{h \in H} \nabla f_r(x_{t',h-1}) \|^2 + \epsilon$. Taking expectation on both sides with respect to the random data samples and re-arranging the terms, we have

$$\mathbb{E}\left[\left\langle -\eta \sum_{h \in H} \nabla f_r(x_{t',h-1}), u \right\rangle\right]$$

$$\geq \rho \mathbb{E} \| \eta \sum_{h \in H} \nabla f_r(x_{t',h-1}) \|^2 + \epsilon$$

$$\geq \rho \| \eta \sum_{h \in H} \nabla F_r(x_{t',h-1}) \|^2 + \epsilon$$

$$\geq \rho \| \eta \sum_{h \in H} [\nabla F_r(x_{t',h-1}) - \nabla F(x_{t',h-1}) + \nabla F(x_{t',h-1})] \|^2 + \epsilon$$

$$\geq \rho \| \eta \sum_{h \in H} [\nabla F_r(x_{t',h-1}) - \nabla F(x_{t',h-1})] \|^2 + \rho \| \eta \sum_{h \in H} \nabla F(x_{t',h-1}) \|^2$$

$$\quad + 2\rho \left\langle \eta \sum_{h \in H} [\nabla F_r(x_{t',h-1}) - \nabla F(x_{t',h-1})], \eta \sum_{h \in H} \nabla F(x_{t',h-1}) \right\rangle + \epsilon$$

$$\geq \rho \| \eta \sum_{h \in H} \nabla F(x_{t',h-1}) \|^2 - 2\rho \eta^2 H^2 r \sqrt{V_3} + \epsilon.$$

On the other hand, we have

$$\mathbb{E}\left[\left\langle -\eta \sum_{h \in H} \nabla f_r(x_{t',h-1}), u \right\rangle\right]$$

$$= \mathbb{E}\left[\left\langle -\eta \sum_{h \in H} \nabla f_r(x_{t',h-1}) + \eta \sum_{h \in H} \nabla F_r(x_{t',h-1}) - \eta \sum_{h \in H} \nabla F_r(x_{t',h-1}), u \right\rangle\right]$$

$$= \mathbb{E}\left[\left\langle -\eta \sum_{h \in H} \nabla F_r(x_{t',h-1}), u \right\rangle\right] + \mathbb{E}\left[\left\langle -\eta \sum_{h \in H} \nabla f_r(x_{t',h-1}) + \eta \sum_{h \in H} \nabla F_r(x_{t',h-1}), u \right\rangle\right]$$

$$\leq \left\langle -\eta \sum_{h \in H} \nabla F_r(x_{t',h-1}), \mathbb{E}[u] \right\rangle + \eta \mathbb{E}\left[\| -\sum_{h \in H} \nabla f_r(x_{t',h-1}) + \sum_{h \in H} \nabla F_r(x_{t',h-1}) \| \times \|u\|\right]$$

$$\leq \left\langle -\eta \sum_{h \in H} \nabla F_r(x_{t',h-1}), \mathbb{E}[u] \right\rangle + \eta^2 H(1+\gamma)\sqrt{V_3} \sum_{h \in H} \mathbb{E}\| -\nabla f_r(x_{t',h-1}) + \nabla F_r(x_{t',h-1}) \|$$

$$\leq \left\langle -\eta \sum_{h \in H} \nabla F_r(x_{t',h-1}), \mathbb{E}[u] \right\rangle + \frac{\eta^2 H^2(1+\gamma)\sqrt{V_3}\sigma}{\sqrt{s_r}}$$

$$= \frac{1}{2} \| -\eta \sum_{h \in H} \nabla F_r(x_{t',h-1}) \|^2 + \frac{1}{2} \|\mathbb{E}[u]\|^2 - \frac{1}{2} \| -\eta \sum_{h \in H} \nabla F_r(x_{t',h-1}) - \mathbb{E}[u] \|^2$$

$$\quad + \frac{\eta^2 H^2(1+\gamma)\sqrt{V_3}\sigma}{\sqrt{s_r}}$$

$$= \frac{1}{2} \| -\eta \sum_{h \in H} \nabla F_r(x_{t',h-1}) \|^2 + \frac{1}{2} \|\mathbb{E}[u]\|^2$$

$$\quad - \frac{1}{2} \| -\eta \sum_{h \in H} [\nabla F_r(x_{t',h-1}) - \nabla F(x_{t',h-1}) + \nabla F(x_{t',h-1})] - \mathbb{E}[u] \|^2 + \frac{\eta^2 H^2(1+\gamma)\sqrt{V_3}\sigma}{\sqrt{s_r}}$$

$$= \frac{1}{2} \| -\eta \sum_{h \in H} \nabla F_r(x_{t',h-1}) \|^2 + \frac{1}{2} \|\mathbb{E}[u]\|^2$$

$$\quad - \frac{1}{2} \| -\eta \sum_{h \in H} [\nabla F_r(x_{t',h-1}) - \nabla F(x_{t',h-1}) + \nabla F(x_{t',h-1})] - \mathbb{E}[u] \|^2 + \frac{\eta^2 H^2(1+\gamma)\sqrt{V_3}\sigma}{\sqrt{s_r}}$$

$$\leq \left\langle -\eta \sum_{h \in H} \nabla F(x_{t',h-1}), \mathbb{E}[u] \right\rangle + \eta^2 H^2 \frac{r^2}{2} + \frac{2+\gamma}{2}\eta^2 H^2 r \sqrt{V_3} + \frac{\eta^2 H^2(1+\gamma)\sqrt{V_3}\sigma}{\sqrt{s_r}}.$$

Combining these, we have

$$
\left\langle -\eta \sum_{h \in H} \nabla F(x_{t',h-1}), \mathbb{E}[u] \right\rangle
$$

$$
\geq \rho \left\| \eta \sum_{h \in H} \nabla F(x_{t',h-1}) \right\|^2 + \epsilon - 2\rho \eta^2 H^2 r \sqrt{V_3} - \eta^2 H^2 \frac{r^2}{2} - \frac{2+\gamma}{2} \eta^2 H^2 r \sqrt{V_3}
$$

$$
- \frac{\eta^2 H^2 (1+\gamma) \sqrt{V_3} \sigma}{\sqrt{s_r}}
$$

$$
\geq 0,
$$

if we take $\epsilon \geq 2\rho \eta^2 H^2 r \sqrt{V_3} + \eta^2 H^2 \frac{r^2}{2} + \frac{2+\gamma}{2} \eta^2 H^2 r \sqrt{V_3} + \frac{\eta^2 H^2 (1+\gamma) \sqrt{V_3} \sigma}{\sqrt{s_r}}$. □

**Theorem A2** (Error bound of ZenoPS in the synchronous mode without compression). *In addition to Assumption 1 (smoothness), Assumption 2 (bounded variance), Assumption 3 (bounded gradients), and Assumption 4 (global minimum), we assume that the compressors are disabled and the validation data is close to the training data $\|\nabla F_r(x) - \nabla F(x)\| \leq r, \forall x \in \mathbb{R}^d$ where $r \geq 0$ and $F_r(x) = \mathbb{E}[f_r(x; z \sim \mathcal{D}_r)]$ (the expectation is taken with respect to z). Taking $\epsilon = cH^2\eta^2$, we have the following error bound for ZenoPS in the synchronous mode:*

$$
\frac{\sum_{t \in [T]} \mathbb{E} \|\nabla F(x_{t-1})\|^2}{T}
$$

$$
\leq \frac{F(x_0) - F(x_*)}{TH\eta\rho} + \frac{3 + 3\gamma + 4\rho}{2\rho} LH\eta V_3 + \frac{(4 + 3\gamma + 4\rho)r\sqrt{V_3} + r^2 - 2c}{2\rho} + \frac{(1+\gamma)\sqrt{V_3}\sigma}{\rho\sqrt{s_r}}.
$$

*Taking $\eta = \frac{1}{\sqrt{TH}}$, $s_r \propto TH$, we have*

$$
\frac{\sum_{t \in [T]} \mathbb{E} \|\nabla F(x_{t-1})\|^2}{T}
$$

$$
\leq \mathcal{O}\left( \frac{F(x_0) - F(x_*)}{\sqrt{TH}\rho} \right) + \mathcal{O}\left( \frac{H}{\sqrt{TH}\rho} \right) + \mathcal{O}\left( \frac{\sigma}{\sqrt{TH}\rho} \right) + \frac{(4 + 3\gamma + 4\rho)r\sqrt{V_3} + r^2 - 2c}{2\rho}.
$$

**Proof.** Using smoothness, we have

$$
F(x_t) - F(x_{t-1})
$$

$$
\leq \langle \nabla F(x_{t-1}), u_t \rangle + \frac{L}{2} \|u_t\|^2
$$

$$
\leq \langle \nabla F(x_{t-1}), u_t \rangle + \frac{L(1+\gamma)}{2} \|v_t\|^2
$$

$$
\leq \langle \nabla F(x_{t-1}), u_t \rangle + \frac{L(1+\gamma)}{2} \eta^2 H^2 V_3,
$$

where $x_t = x_{t-1} + u_t$, $v_t$ is the validation vector. Taking the expectation on both sides, we have

$$\mathbb{E}[F(x_t) - F(x_{t-1})]$$

$$\leq \langle \nabla F(x_{t-1}), \mathbb{E}[u_t] \rangle + \frac{L(1+\gamma)}{2} \eta^2 H^2 V_3$$

$$\leq \frac{1}{H\eta} \langle H\eta \nabla F(x_{t-1}), \mathbb{E}[u_t] \rangle + \frac{L(1+\gamma)}{2} \eta^2 H^2 V_3$$

$$\leq \frac{1}{H} \eta \langle H\eta \nabla F(x_{t-1}) - \mathbb{E}[v_t] + \mathbb{E}[v_t], \mathbb{E}[u_t] \rangle + \frac{L(1+\gamma)}{2} \eta^2 H^2 V_3$$

$$\leq -\frac{1}{H\eta} \langle \mathbb{E}[v_t], \mathbb{E}[u_t] \rangle + \frac{1}{H\eta} \langle H\eta \nabla F(x_{t-1}) + \mathbb{E}[v_t], \mathbb{E}[u_t] \rangle + \frac{L(1+\gamma)}{2} \eta^2 H^2 V_3.$$

Using the results in the proof of Byzantine tolerance, we have

$$\mathbb{E}[F(x_t) - F(x_{t-1})]$$

$$\leq -\frac{1}{H\eta} \langle \mathbb{E}[v_t], \mathbb{E}[u_t] \rangle + \frac{1}{H\eta} \langle H\eta \nabla F(x_{t-1}) + \mathbb{E}[v_t], \mathbb{E}[u_t] \rangle + \frac{L(1+\gamma)}{2} \eta^2 H^2 V_3$$

$$\leq -\frac{1}{H\eta} \rho \| \eta \sum_{h \in H} \nabla F(x_{t',h-1}) \|^2$$

$$\quad + \frac{1}{H\eta} [2\rho \eta^2 H^2 r \sqrt{V_3} + \eta^2 H^2 \frac{r^2}{2} + \frac{2+\gamma}{2} \eta^2 H^2 r \sqrt{V_3} + \frac{\eta^2 H^2 (1+\gamma) \sqrt{V_3} \sigma}{\sqrt{s_r}} - \epsilon]$$

$$\quad + \frac{1}{H\eta} \langle H\eta \nabla F(x_{t-1}) + \mathbb{E}[v_t], \mathbb{E}[u_t] \rangle + \frac{L(1+\gamma)}{2} \eta^2 H^2 V_3$$

$$\leq -\frac{\rho}{H\eta} \| \eta \sum_{h \in H} \nabla F(x_{t',h-1}) \|^2 + \eta H \frac{r^2}{2} + \frac{2+\gamma}{2} \eta H r \sqrt{V_3} + \frac{\eta H (1+\gamma) \sqrt{V_3} \sigma}{\sqrt{s_r}}$$

$$\quad + \frac{1}{H\eta} \langle H\eta \nabla F(x_{t-1}) + \mathbb{E}[v_t], \mathbb{E}[u_t] \rangle + \frac{L(1+\gamma)}{2} \eta^2 H^2 V_3 + 2\rho \eta H r \sqrt{V_3} - \frac{\epsilon}{H\eta}$$

$$\leq -\frac{\rho}{H\eta} \| \eta \sum_{h \in H} [\nabla F(x_{t',h-1}) - \nabla F(x_{t-1}) + \nabla F(x_{t-1})] \|^2$$

$$\quad + \frac{1}{H\eta} \langle H\eta \nabla F(x_{t-1}) + \mathbb{E}[v_t], \mathbb{E}[u_t] \rangle + \frac{L(1+\gamma)}{2} \eta^2 H^2 V_3 + 2\rho \eta H r \sqrt{V_3} - \frac{\epsilon}{H\eta}$$

$$\quad + \eta H \frac{r^2}{2} + \frac{2+\gamma}{2} \eta H r \sqrt{V_3} + \frac{\eta H (1+\gamma) \sqrt{V_3} \sigma}{\sqrt{s_r}}$$

$$\leq -\frac{\rho}{H\eta} \| \eta \sum_{h \in H} \nabla F(x_{t-1}) \|^2 + \frac{2\rho}{H\eta} \left\| \eta \sum_{h \in H} [\nabla F(x_{t',h-1}) - \nabla F(x_{t-1})] \right\| \times \left\| \eta \sum_{h \in H} \nabla F(x_{t-1}) \right\|$$

$$\quad + \frac{1}{H\eta} \langle H\eta \nabla F(x_{t-1}) + \mathbb{E}[v_t], \mathbb{E}[u_t] \rangle + \frac{L(1+\gamma)}{2} \eta^2 H^2 V_3 + 2\rho \eta H r \sqrt{V_3} - \frac{\epsilon}{H\eta}$$

$$\quad + \eta H \frac{r^2}{2} + \frac{2+\gamma}{2} \eta H r \sqrt{V_3} + \frac{\eta H (1+\gamma) \sqrt{V_3} \sigma}{\sqrt{s_r}}$$

$$\leq -H\eta \rho \| \nabla F(x_{t-1}) \|^2 + 2\rho \eta \sqrt{V_3} \left\| \sum_{h \in H} [\nabla F(x_{t',h-1}) - \nabla F(x_{t-1})] \right\|$$

$$\quad + \frac{1}{H\eta} \langle H\eta \nabla F(x_{t-1}) + \mathbb{E}[v_t], \mathbb{E}[u_t] \rangle + \frac{L(1+\gamma)}{2} \eta^2 H^2 V_3 + 2\rho \eta H r \sqrt{V_3} - \frac{\epsilon}{H\eta}$$

$$\quad + \eta H \frac{r^2}{2} + \frac{2+\gamma}{2} \eta H r \sqrt{V_3} + \frac{\eta H (1+\gamma) \sqrt{V_3} \sigma}{\sqrt{s_r}},$$

where $t' = t - 1$.

It is easy to check that

$$\left\| \sum_{h \in H} [\nabla F(x_{t',h-1}) - \nabla F(x_{t-1})] \right\|$$
$$\le \sum_{h \in H} \|\nabla F(x_{t',h-1}) - \nabla F(x_{t-1})\|$$
$$\le \sum_{h \in H} L \|x_{t',h-1} - x_{t-1}\|$$
$$\le L H^2 \eta \sqrt{V_3}.$$

Thus, we have

$$\mathbb{E}[F(x_t) - F(x_{t-1})]$$

$$\le -H\eta\rho\|\nabla F(x_{t-1})\|^2 + 2\rho\eta\sqrt{V_3} \left\| \sum_{h \in H} [\nabla F(x_{t',h-1}) - \nabla F(x_{t-1})] \right\|$$

$$\quad + \frac{1}{H\eta}\langle H\eta\nabla F(x_{t-1}) + \mathbb{E}[v_t], \mathbb{E}[u_t]\rangle + \frac{L(1+\gamma)}{2}\eta^2 H^2 V_3 + 2\rho\eta Hr\sqrt{V_3} - \frac{\epsilon}{H\eta}$$

$$\quad + \eta H \frac{r^2}{2} + \frac{2+\gamma}{2}\eta Hr\sqrt{V_3} + \frac{\eta H(1+\gamma)\sqrt{V_3}\sigma}{\sqrt{s_r}}$$

$$\le -H\eta\rho\|\nabla F(x_{t-1})\|^2 + 2\rho L H^2 \eta^2 V_3$$

$$\quad + \frac{1}{H\eta}\langle H\eta\nabla F(x_{t-1}) + \mathbb{E}[v_t], \mathbb{E}[u_t]\rangle + \frac{L(1+\gamma)}{2}\eta^2 H^2 V_3 + 2\rho\eta Hr\sqrt{V_3} - \frac{\epsilon}{H\eta}$$

$$\quad + \eta H \frac{r^2}{2} + \frac{2+\gamma}{2}\eta Hr\sqrt{V_3} + \frac{\eta H(1+\gamma)\sqrt{V_3}\sigma}{\sqrt{s_r}}$$

$$\le -H\eta\rho\|\nabla F(x_{t-1})\|^2 + \frac{1}{H\eta}\|H\eta\nabla F(x_{t-1}) + \mathbb{E}[v_t]\| \times \|\mathbb{E}[u_t]\|$$

$$\quad + \frac{L(1+\gamma)}{2}\eta^2 H^2 V_3 + 2\rho\eta Hr\sqrt{V_3} - \frac{\epsilon}{H\eta} + 2\rho L H^2 \eta^2 V_3$$

$$\quad + \eta H \frac{r^2}{2} + \frac{2+\gamma}{2}\eta Hr\sqrt{V_3} + \frac{\eta H(1+\gamma)\sqrt{V_3}\sigma}{\sqrt{s_r}}.$$

To finish the upper bound, we have

$$\|H\eta\nabla F(x_{t-1}) + \mathbb{E}[v_t]\|$$
$$= \left\| \sum_{h \in [H]} \eta[\nabla F(x_{t-1}) - \nabla F_r(x_{t-1,h-1})] \right\|$$
$$\le \eta \sum_{h \in [H]} \|\nabla F(x_{t-1}) - \nabla F_r(x_{t-1,h-1})\|$$
$$\le \eta \sum_{h \in [H]} [\|\nabla F(x_{t-1}) - \nabla F(x_{t-1,h-1})\| + r]$$
$$\le \eta \sum_{h \in [H]} [L\|x_{t-1} - x_{t-1,h-1}\| + r]$$
$$\le \eta \sum_{h \in [H]} [LH\eta\sqrt{V_3} + r]$$
$$\le \eta H[LH\eta\sqrt{V_3} + r].$$

On the other hand, we have

$$\|\mathbb{E}[u_t]\|$$
$$\leq \mathbb{E}\|u_t\|$$
$$\leq (1+\gamma)\mathbb{E}\|v_t\|$$
$$\leq (1+\gamma)H\eta\sqrt{V_3}.$$

Thus, we have

$$\mathbb{E}[F(x_t) - F(x_{t-1})]$$
$$\leq -H\eta\rho\|\nabla F(x_{t-1})\|^2 + \frac{1}{H\eta}\|H\eta\nabla F(x_{t-1}) + \mathbb{E}[v_t]\| \times \|\mathbb{E}[u_t]\|$$
$$+ \frac{L(1+\gamma)}{2}\eta^2 H^2 V_3 + 2\rho\eta Hr\sqrt{V_3} - \frac{\epsilon}{H\eta} + 2\rho LH^2\eta^2 V_3$$
$$+ \eta H\frac{r^2}{2} + \frac{2+\gamma}{2}\eta Hr\sqrt{V_3} + \frac{\eta H(1+\gamma)\sqrt{V_3}\sigma}{\sqrt{s_r}}$$
$$\leq -H\eta\rho\|\nabla F(x_{t-1})\|^2 + (1+\gamma)LH^2\eta^2 V_3 + (1+\gamma)H\eta r\sqrt{V_3}$$
$$+ \frac{L(1+\gamma)}{2}\eta^2 H^2 V_3 + 2\rho\eta Hr\sqrt{V_3} - \frac{\epsilon}{H\eta} + 2\rho LH^2\eta^2 V_3$$
$$+ \eta H\frac{r^2}{2} + \frac{2+\gamma}{2}\eta Hr\sqrt{V_3} + \frac{\eta H(1+\gamma)\sqrt{V_3}\sigma}{\sqrt{s_r}}.$$

By re-arranging the terms, telescoping, and taking the total expectation, we have

$$\frac{\sum_{t\in[T]}\mathbb{E}\|\nabla F(x_{t-1})\|^2}{T}$$
$$\leq \frac{\mathbb{E}[F(x_0) - F(x_T)]}{TH\eta\rho} + \frac{3+3\gamma+4\rho}{2\rho}LH\eta V_3 + \frac{4+3\gamma+4\rho}{2\rho}r\sqrt{V_3} + \frac{r^2}{2\rho} - \frac{\epsilon}{H^2\eta^2\rho}$$
$$+ \frac{(1+\gamma)\sqrt{V_3}\sigma}{\rho\sqrt{s_r}},$$

which concludes the proof. □

**Theorem A3** (Error bound of ZenoPS in the synchronous mode with compression). *In addition to Assumption 1 (smoothness), Assumption 3 (bounded gradients), and Assumption 4 (global minimum), we assume that the compressors are disabled and the validation data is close to the training data* $\|\nabla F_r(x) - \nabla F(x)\| \leq r, \forall x \in \mathbb{R}^d$ *where* $r \geq 0$ *and* $F_r(x) = \mathbb{E}[f_r(x; z \sim \mathcal{D}_r)]$ *(the expectation is taken with respect to z). Taking* $\epsilon = cH^2\eta^2$ *and* $\eta = \frac{1}{\sqrt{TH}}$*, we have the following error bound for ZenoPS in the synchronous mode:*

$$\frac{\sum_{t\in[T]}\mathbb{E}\|\nabla F(x_{t-1})\|^2}{T}$$
$$\leq \mathcal{O}\left(\frac{F(x_0) - F(x_*)}{\sqrt{TH}\rho}\right) + \mathcal{O}\left(\frac{H}{\sqrt{TH}\rho}\right) + \mathcal{O}\left(\frac{\sigma}{\sqrt{TH}\rho}\right) + \frac{(4+3\gamma+4\rho)r\sqrt{V_3}+r^2-2c}{2\rho}$$
$$+ \frac{(1-\delta)(1+\gamma)L\sqrt{H}V_3}{2\rho\sqrt{T}\left(1-\sqrt{1-\delta}\right)^2}.$$

**Proof.** We only need to add the additional error caused by $e_{i,t}$ to the previous error bound. We already have $e_{i,0} = 0$. For any $t \geq 1$, we can bound the local error:

$$\mathbb{E}\|e_{i,t}\|^2$$

$$= (1-\delta)\mathbb{E}\left\|e_{i,t-1} - \eta \sum_{h\in[H]} g_{i,t-1,h-1}\right\|^2$$

$$\leq (1-\delta)(1+a)\mathbb{E}\|e_{i,t-1}\|^2 + (1-\delta)(1+1/a)\mathbb{E}\left\|\eta \sum_{h\in[H]} g_{i,t-1,h-1}\right\|^2$$

$$\leq (1-\delta)(1+a)\mathbb{E}\|e_{i,t-1}\|^2 + (1-\delta)\eta^2(1+1/a)\mathbb{E}\left\|\sum_{h\in[H]} g_{i,t-1,h-1}\right\|^2$$

$$\leq (1-\delta)(1+a)\mathbb{E}\|e_{i,t-1}\|^2 + (1-\delta)H^2\eta^2(1+1/a)V_3$$

$$\leq (1+1/a)(1-\delta)H^2\eta^2 V_3 \sum_{t'=0}^{+\infty}[(1+a)(1-\delta)]^{t'}$$

$$\leq \frac{1+1/a}{1-(1+a)(1-\delta)}(1-\delta)H^2\eta^2 V_3,$$

for any $a > 0$, such that $(1+a)(1-\delta_1) \in (0,1)$. The bound above is minimized when we take $a = \frac{1}{\sqrt{1-\delta}} - 1$, which results in

$$\mathbb{E}\|e_{i,t}\|^2 \leq \frac{(1-\delta)H^2\eta^2 V_3}{\left(1-\sqrt{1-\delta}\right)^2}.$$

$\square$

**Theorem A4** (Error bound of ZenoPS in the asynchronous model with compression)**.** *In addition to Assumption 1 (smoothness), Assumption 2 (bounded variance), Assumption 3 (bounded gradients), and Assumption 4 (global minimum), we assume that the validation data is close to the training data $\|\nabla F_r(x) - \nabla F(x)\| \leq r, \forall x \in \mathbb{R}^d$ where $r \geq 0$ and $F_r(x) = \mathbb{E}[f_r(x; z \sim \mathcal{D}_r)]$ (the expectation is taken with respect to z). Furthermore, we assume that, in any server step t, the approved update is based on the global model parameters in the server step $t'$, where $t' \leq t-1$ has bounded delay $t - 1 - t' \leq \tau$. Taking $\epsilon = cH^2\eta^2$, we have the following error bound for ZenoPS in the asynchronous mode:*

$$\frac{\sum_{t\in[T]} \mathbb{E}\|\nabla F(x_{t-1})\|^2}{T}$$

$$\leq \frac{F(x_0) - F(x_*)}{TH\alpha\eta\rho} + \frac{(2\tau+2+\alpha)(1+\gamma) + 4(\tau+1)\rho}{\rho}LH\eta V_3$$

$$+ \frac{(2+\gamma+4\rho)r\sqrt{V_3} + r^2 - 2c}{2\rho} + \frac{(1+\gamma)\sqrt{V_3}\sigma}{\rho\sqrt{s_r}} + \frac{(1-\delta)(1+\gamma)L\sqrt{H}V_3}{2\rho\sqrt{T}\left(1-\sqrt{1-\delta}\right)^2}.$$

*Taking $\eta = \frac{1}{\sqrt{TH}}$, $s_r \propto TH$, we have*

$$\frac{\sum_{t\in[T]} \mathbb{E}\|\nabla F(x_{t-1})\|^2}{T}$$

$$\leq \frac{F(x_0) - F(x_*)}{\sqrt{TH}\alpha\rho} + \sqrt{\frac{H}{T}}\frac{(2\tau+2+\alpha)(1+\gamma) + 4(\tau+1)\rho}{\rho}LV_3$$

$$+ \frac{(2+\gamma+4\rho)r\sqrt{V_3} + r^2 - 2c}{2\rho} + \frac{(1+\gamma)\sqrt{V_3}\sigma}{\rho\sqrt{TH}} + \frac{(1-\delta)(1+\gamma)L\sqrt{H}V_3}{2\rho\sqrt{T}\left(1-\sqrt{1-\delta}\right)^2}.$$

**Proof.** Using smoothness, we have

$$F(x_t) - F(x_{t-1})$$

$$\leq \langle \nabla F(x_{t-1}), \alpha u_t \rangle + \frac{L\alpha^2}{2} \|u_t\|^2$$

$$\leq \alpha \langle \nabla F(x_{t-1}), \alpha u_t \rangle + \frac{L\alpha^2(1+\gamma)}{2} \|v_t\|^2$$

$$\leq \alpha \langle \nabla F(x_{t-1}), u_t \rangle + \frac{L\alpha^2(1+\gamma)}{2} \eta^2 H^2 V_3,$$

where $x_t = x_{t-1} + u_t$, and $v_t$ is the validation vector. Taking the expectation on both sides, we have

$$\mathbb{E}[F(x_t) - F(x_{t-1})]$$

$$\leq \alpha \langle \nabla F(x_{t-1}), \mathbb{E}[u_t] \rangle + \frac{L\alpha^2(1+\gamma)}{2} \eta^2 H^2 V_3$$

$$\leq \frac{\alpha}{H\eta} \langle H\eta \nabla F(x_{t-1}), \mathbb{E}[u_t] \rangle + \frac{L\alpha^2(1+\gamma)}{2} \eta^2 H^2 V_3$$

$$\leq \frac{\alpha}{H} \eta \langle H\eta \nabla F(x_{t-1}) - \mathbb{E}[v_t] + \mathbb{E}[v_t], \mathbb{E}[u_t] \rangle + \frac{L\alpha^2(1+\gamma)}{2} \eta^2 H^2 V_3$$

$$\leq -\frac{\alpha}{H\eta} \langle \mathbb{E}[v_t], \mathbb{E}[u_t] \rangle + \frac{\alpha}{H\eta} \langle H\eta \nabla F(x_{t-1}) + \mathbb{E}[v_t], \mathbb{E}[u_t] \rangle + \frac{L\alpha^2(1+\gamma)}{2} \eta^2 H^2 V_3.$$

Using the results in the proof of Byzantine tolerance, we have

$$\mathbb{E}[F(x_t) - F(x_{t-1})]$$

$$\leq -\frac{\alpha}{H\eta} \langle \mathbb{E}[v_t], \mathbb{E}[u_t] \rangle + \frac{\alpha}{H\eta} \langle H\eta \nabla F(x_{t-1}) + \mathbb{E}[v_t], \mathbb{E}[u_t] \rangle + \frac{L\alpha^2(1+\gamma)}{2} \eta^2 H^2 V_3$$

$$\leq -\frac{\alpha}{H\eta} \rho \|\eta \sum_{h \in H} \nabla F(x_{t',h-1})\|^2$$

$$+ \frac{\alpha}{H\eta} [2\rho \eta^2 H^2 r \sqrt{V_3} + \eta^2 H^2 \frac{r^2}{2} + \frac{2+\gamma}{2} \eta^2 H^2 r \sqrt{V_3} + \frac{\eta^2 H^2 (1+\gamma) \sqrt{V_3} \sigma}{\sqrt{s_r}} - \epsilon]$$

$$+ \frac{\alpha}{H\eta} \langle H\eta \nabla F(x_{t-1}) + \mathbb{E}[v_t], \mathbb{E}[u_t] \rangle + \frac{L\alpha^2(1+\gamma)}{2} \eta^2 H^2 V_3$$

$$\leq -\frac{\alpha\rho}{H\eta} \|\eta \sum_{h \in H} \nabla F(x_{t',h-1})\|^2$$

$$+ \frac{\alpha}{H\eta} \langle H\eta \nabla F(x_{t-1}) + \mathbb{E}[v_t], \mathbb{E}[u_t] \rangle + \frac{L\alpha^2(1+\gamma)}{2} \eta^2 H^2 V_3$$

$$+ \frac{\alpha}{H\eta} [2\rho \eta^2 H^2 r \sqrt{V_3} + \eta^2 H^2 \frac{r^2}{2} + \frac{2+\gamma}{2} \eta^2 H^2 r \sqrt{V_3} + \frac{\eta^2 H^2 (1+\gamma) \sqrt{V_3} \sigma}{\sqrt{s_r}} - \epsilon]$$

$$\leq -\frac{\alpha\rho}{H\eta} \|\eta \sum_{h \in H} [\nabla F(x_{t',h-1}) - \nabla F(x_{t-1}) + \nabla F(x_{t-1})]\|^2$$

$$+ \frac{\alpha}{H\eta} \langle H\eta \nabla F(x_{t-1}) + \mathbb{E}[v_t], \mathbb{E}[u_t] \rangle + \frac{L\alpha^2(1+\gamma)}{2} \eta^2 H^2 V_3$$

$$+ \frac{\alpha}{H\eta} [2\rho \eta^2 H^2 r \sqrt{V_3} + \eta^2 H^2 \frac{r^2}{2} + \frac{2+\gamma}{2} \eta^2 H^2 r \sqrt{V_3} + \frac{\eta^2 H^2 (1+\gamma) \sqrt{V_3} \sigma}{\sqrt{s_r}} - \epsilon]$$

$$\leq -\frac{\alpha\rho}{H\eta} \|\eta \sum_{h \in H} \nabla F(x_{t-1})\|^2 + \frac{2\alpha\rho}{H\eta} \left\|\eta \sum_{h \in H} [\nabla F(x_{t',h-1}) - \nabla F(x_{t-1})]\right\| \times \left\|\eta \sum_{h \in H} \nabla F(x_{t-1})\right\|$$

$$+ \frac{\alpha}{H\eta}\langle H\eta\nabla F(x_{t-1}) + \mathbb{E}[v_t], \mathbb{E}[u_t]\rangle + \frac{L\alpha^2(1+\gamma)}{2}\eta^2 H^2 V_3$$

$$+ \frac{\alpha}{H\eta}[2\rho\eta^2 H^2 r\sqrt{V_3} + \eta^2 H^2\frac{r^2}{2} + \frac{2+\gamma}{2}\eta^2 H^2 r\sqrt{V_3} + \frac{\eta^2 H^2(1+\gamma)\sqrt{V_3}\sigma}{\sqrt{s_r}} - \epsilon]$$

$$\leq -H\alpha\eta\rho\|\nabla F(x_{t-1})\|^2 + 2\alpha\rho\eta\sqrt{V_3}\left\|\sum_{h\in H}[\nabla F(x_{t',h-1}) - \nabla F(x_{t-1})]\right\|$$

$$+ \frac{\alpha}{H\eta}\langle H\eta\nabla F(x_{t-1}) + \mathbb{E}[v_t], \mathbb{E}[u_t]\rangle + \frac{L\alpha^2(1+\gamma)}{2}\eta^2 H^2 V_3$$

$$+ \frac{\alpha}{H\eta}[2\rho\eta^2 H^2 r\sqrt{V_3} + \eta^2 H^2\frac{r^2}{2} + \frac{2+\gamma}{2}\eta^2 H^2 r\sqrt{V_3} + \frac{\eta^2 H^2(1+\gamma)\sqrt{V_3}\sigma}{\sqrt{s_r}} - \epsilon],$$

where $t' \leq t - 1$.

Using $(t - 1) - t' \leq \tau$, we have

$$\left\|\sum_{h\in H}[\nabla F(x_{t',h-1}) - \nabla F(x_{t-1})]\right\|$$

$$\leq \sum_{h\in H}\|\nabla F(x_{t',h-1}) - \nabla F(x_{t-1})\|$$

$$\leq \sum_{h\in H}L\|x_{t',h-1} - x_{t-1}\|$$

$$\leq \sum_{h\in H}L[\|x_{t',h-1} - x_{t'}\| + \|x_{t'} - x_{t-1}\|]$$

$$\leq 2(\tau+1)LH^2\eta\sqrt{V_3}.$$

Thus, we have

$$\mathbb{E}[F(x_t) - F(x_{t-1})]$$

$$\leq -H\alpha\eta\rho\|\nabla F(x_{t-1})\|^2 + 2\alpha\rho\eta\sqrt{V_3}\left\|\sum_{h\in H}[\nabla F(x_{t',h-1}) - \nabla F(x_{t-1})]\right\|$$

$$+ \frac{\alpha}{H\eta}\langle H\eta\nabla F(x_{t-1}) + \mathbb{E}[v_t], \mathbb{E}[u_t]\rangle + \frac{L\alpha^2(1+\gamma)}{2}\eta^2 H^2 V_3$$

$$+ \frac{\alpha}{H\eta}[2\rho\eta^2 H^2 r\sqrt{V_3} + \eta^2 H^2\frac{r^2}{2} + \frac{2+\gamma}{2}\eta^2 H^2 r\sqrt{V_3} + \frac{\eta^2 H^2(1+\gamma)\sqrt{V_3}\sigma}{\sqrt{s_r}} - \epsilon]$$

$$\leq -H\alpha\eta\rho\|\nabla F(x_{t-1})\|^2 + 4\alpha(\tau+1)\rho LH^2\eta^2 V_3$$

$$+ \frac{\alpha}{H\eta}\langle H\eta\nabla F(x_{t-1}) + \mathbb{E}[v_t], \mathbb{E}[u_t]\rangle + \frac{L\alpha^2(1+\gamma)}{2}\eta^2 H^2 V_3$$

$$+ \frac{\alpha}{H\eta}[2\rho\eta^2 H^2 r\sqrt{V_3} + \eta^2 H^2\frac{r^2}{2} + \frac{2+\gamma}{2}\eta^2 H^2 r\sqrt{V_3} + \frac{\eta^2 H^2(1+\gamma)\sqrt{V_3}\sigma}{\sqrt{s_r}} - \epsilon]$$

$$\leq -H\alpha\eta\rho\|\nabla F(x_{t-1})\|^2 + \frac{\alpha}{H\eta}\|H\eta\nabla F(x_{t-1}) + \mathbb{E}[v_t]\| \times \|\mathbb{E}[u_t]\|$$

$$+ \frac{L\alpha^2(1+\gamma)}{2}\eta^2 H^2 V_3 + 4\alpha(\tau+1)\rho LH^2\eta^2 V_3$$

$$+ \frac{\alpha}{H\eta}[2\rho\eta^2 H^2 r\sqrt{V_3} + \eta^2 H^2\frac{r^2}{2} + \frac{2+\gamma}{2}\eta^2 H^2 r\sqrt{V_3} + \frac{\eta^2 H^2(1+\gamma)\sqrt{V_3}\sigma}{\sqrt{s_r}} - \epsilon].$$

To finish the upper bound, we have

$$\|H\eta\nabla F(x_{t-1}) + \mathbb{E}[v_t]\|$$
$$= \|\sum_{h\in[H]} \eta[\nabla F(x_{t-1}) - \nabla F_r(x_{t',h-1})]\|$$
$$\leq \eta \sum_{h\in[H]} \|\nabla F(x_{t-1}) - \nabla F_r(x_{t',h-1})\|$$
$$\leq \eta \sum_{h\in[H]} [\|\nabla F(x_{t-1}) - \nabla F(x_{t',h-1})\| + r]$$
$$\leq \eta \sum_{h\in[H]} [L\|x_{t-1} - x_{t',h-1}\| + r]$$
$$\leq \eta \sum_{h\in[H]} [2(\tau+1)LH\eta\sqrt{V_3} + r]$$
$$\leq \eta H[2(\tau+1)LH\eta\sqrt{V_3} + r].$$

On the other hand, we have

$$\|\mathbb{E}[u_t]\|$$
$$\leq \mathbb{E}\|u_t\|$$
$$\leq (1+\gamma)\mathbb{E}\|v_t\|$$
$$\leq (1+\gamma)H\eta\sqrt{V_3}.$$

Thus, we have

$$\mathbb{E}[F(x_t) - F(x_{t-1})]$$
$$\leq -H\alpha\eta\rho\|\nabla F(x_{t-1})\|^2 + \frac{\alpha}{H\eta}\|H\eta\nabla F(x_{t-1}) + \mathbb{E}[v_t]\| \times \|\mathbb{E}[u_t]\|$$
$$+ \frac{L\alpha^2(1+\gamma)}{2}\eta^2 H^2 V_3 + 4\alpha(\tau+1)\rho LH^2\eta^2 V_3$$
$$+ \frac{\alpha}{H\eta}[2\rho\eta^2 H^2 r\sqrt{V_3} + \eta^2 H^2\frac{r^2}{2} + \frac{2+\gamma}{2}\eta^2 H^2 r\sqrt{V_3} + \frac{\eta^2 H^2(1+\gamma)\sqrt{V_3}\sigma}{\sqrt{s_r}} - \epsilon]$$
$$\leq -H\alpha\eta\rho\|\nabla F(x_{t-1})\|^2 + 2\alpha(\tau+1)(1+\gamma)LH^2\eta^2 V_3 + \alpha(1+\gamma)H\eta r\sqrt{V_3}$$
$$+ \frac{L\alpha^2(1+\gamma)}{2}\eta^2 H^2 V_3 + 4\alpha(\tau+1)\rho LH^2\eta^2 V_3$$
$$+ \frac{\alpha}{H\eta}[2\rho\eta^2 H^2 r\sqrt{V_3} + \eta^2 H^2\frac{r^2}{2} + \frac{2+\gamma}{2}\eta^2 H^2 r\sqrt{V_3} + \frac{\eta^2 H^2(1+\gamma)\sqrt{V_3}\sigma}{\sqrt{s_r}} - \epsilon].$$

By re-arranging the terms, telescoping, and taking the total expectation, we have

$$\frac{\sum_{t\in[T]} \mathbb{E}\|\nabla F(x_{t-1})\|^2}{T}$$
$$\leq \frac{\mathbb{E}[F(x_0) - F(x_T)]}{TH\alpha\eta\rho} + \frac{(2\tau+2+\alpha)(1+\gamma) + 4(\tau+1)\rho}{\rho}LH\eta V_3$$
$$+ \frac{(2+\gamma+4\rho)r\sqrt{V_3} + r^2}{2\rho} - \frac{\epsilon}{H^2\eta^2\rho} + \frac{(1+\gamma)\sqrt{V_3}\sigma}{\rho\sqrt{s_r}}.$$

Similar to the synchronous mode, we then add the additional compression error to obtain the final result.  □

**Theorem A5** (LDP of the Byzantine mechanism). *Assume that the noise distribution $\mathcal{D}_{noise}$ has the support $[a, b]$, and $p_- = \min_{z\in[a,b]} p_{noise}(z) > 0$, where $p_{noise}(\cdot)$ is the PDF of the noise distribution $\mathcal{D}_{noise}$ (e.g., uniform distribution with support $[a, b]$). The Byzantine mechanism is $\xi$-LDP, where*

$$\xi = \frac{1 - p_{byz}}{p_{byz} p_-}.$$

**Proof.** Denote $q_v(z)$ as the probability density function of the output of the Byzantine mechanism, given the input $v$. For an arbitrary output $z \in Range(\mathcal{M}_{byz})$, we have

$$q_v(z) = \mathbf{1}\{\mathcal{M}_{byz}(v) = z\}[(1 - p_{byz})\mathbf{1}\{v = z\} + p_{byz}p_{noise}(z)] + \mathbf{1}\{\mathcal{M}_{byz}(v) \neq z\}p_{noise}(z). \tag{A1}$$

Using $0 \leq \mathbf{1}\{v = z\} \leq 1$, we have

$$
\begin{aligned}
&q_v(z) \\
&\leq \mathbf{1}\{\mathcal{M}_{byz}(v) = z\}[(1 - p_{byz})(1 - p_{noise}(z))] + p_{noise}(z) \\
&\leq (1 - p_{byz})(1 - p_{noise}(z)) + p_{noise}(z),
\end{aligned}
$$

and

$$
\begin{aligned}
&q_v(z) \\
&\geq \mathbf{1}\{\mathcal{M}_{byz}(v) = z\}p_{byz}p_{noise}(z) + \mathbf{1}\{\mathcal{M}_{byz}(v) \neq z\}p_{noise}(z) \\
&\geq \mathbf{1}\{\mathcal{M}_{byz}(v) = z\}p_{byz}p_{noise}(z) + \mathbf{1}\{\mathcal{M}_{byz}(v) \neq z\}p_{byz}p_{noise}(z) \\
&\geq p_{byz}p_{noise}(z).
\end{aligned}
$$

Thus, for any pair of inputs $v, v'$, we have

$$
\begin{aligned}
&\frac{q_v(z)}{q_{v'}(z)} \\
&\leq \frac{(1 - p_{byz})(1 - p_{noise}(z)) + p_{noise}(z)}{p_{byz}p_{noise}(z)} \\
&= \frac{(1 - p_{byz}) + p_{byz}p_{noise}(z)}{p_{byz}p_{noise}(z)} \\
&\leq \exp\left(\frac{1 - p_{byz}}{p_{byz}p_{noise}(z)}\right) \qquad\qquad \triangleright\, 1 + x \leq \exp(x) \\
&\leq \exp\left(\frac{1 - p_{byz}}{p_{byz}p_-}\right),
\end{aligned}
$$

which concludes the proof. □

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
