# Peer review of "ZenoPS: A Distributed Learning System Integrating Communication Efficiency and Security"

_algorithms, doi:10.3390/a15070233_

Round 1

Reviewer 1 Report

This paper introduced a variant of Stochastic Gradient Descent with improved communication efficiency and security in distributed environments. The proposed prototype of the distributed learning system, ZenoPS, integrates message compression, infrequent synchronization, both asynchronous and synchronous training, and score-based validation. The paper is interesting and well organized. My comments are listed as following,

1. Why the authors used two kinds of attacks( Sign-flipping attack, Random attack) to validate the system.

2. When changing the number of worker workers, any model parameters need to be adjusted to remain the efficiency?

3. How to optimized the hyperparameter ρ?

4. If the workers have different capabilities and communication speeds, how will they affect the system.

5. Conclusion section should be rearranged. According to the topic of the paper, the authors may propose some interesting problems as future work in the conclusion.

This study may be proposed for publication if it is addressed in the specified problems.

Author Response

We thank the reviewer for the detailed comments.

Please refer to the attached pdf file for the detailed responses.

Reviewer 2 Report

1.     Put a paragraph which shows the structure of this manuscript at the end of the introduction section.

2.     In related section, you should describe the pros and cons of each referred papers.

3.     Check all statements of the manuscript. There are some mis-typo such as “the parameter parameter-server (PS)” at the line of 140.

4.     Do you have any reasons that you set parameters with fixed values at section 6.1. Describe the reasons. For example, the reasons can be that the values are selected in order to have the best results after several experiments compared with other methods.

5.      Analyze the reason why the results are differently varied at 100 epochs in the figure 3 and 4.

Author Response

(The authors gave the same response as above.)

Reviewer 3 Report

This paper proposed a distributed learning system for performing machine learning tasks on multiple devices. A variant of stochastic gradient descent approach integrated with error reset is adapted in application. A score-based validation is conducted which confirms the efficiency and feasibility of the proposed system. Overall, this is a well-written paper and technically sound. The reviewer would suggest some revisions before publication:

First, the Byzantine failures needs to be introduced with more explanatory examples. Please provide few examples in real life explaining Byzantine failures.

Second, in the energy sector, the proposed distributed system can benefit multi-sensor based machine learning systems. Please cite the following two paper and discuss how the proposed system can be applied in the energy sector:

Li, H., Deng, J., Feng, P., Pu, C., Arachchige, D. D., & Cheng, Q. (2021). Short-Term Nacelle Orientation Forecasting Using Bilinear Transformation and ICEEMDAN Framework. Frontiers in Energy Research, 697.

Li, H. SCADA Data based Wind Power Interval Prediction using LUBE-based Deep Residual Networks. Frontiers in Energy Research, 690.

Third, during the machine-learning process, how the proposed approach would handle gradient vanishing scenarios? Please give more details.

Last, in the introduction section, please list a few potential in-field applications that the ZenoPS can be applied on. This would help the readers the potentiality of the proposed distributed learning system.

Overall, this is a good research and can be considered for publication after revision.

Author Response

(The authors gave the same response as above.)

Round 2

Reviewer 2 Report

The revision made by the authors have improved the manuscript.